

# Estimating spatiotemporally continuous snow water equivalent from intermittent satellite track observations using machine learning methods

Xiaoyu Ma[1], Dongyue Li[1,2,*], Yiwen Fang[2], Steven Margulis[2], Dennis P. Lettenmaier[1,2]

1Department of Geography, University of California, Los Angeles, 90095, United States
2Department of Civil & Environmental Engineering, University of California, Los Angeles, 90095, United States

* Correspondence to: Dongyue Li (dongyueli@ucla.edu)

**Abstract.** Accurate remote sensing-based snow water equivalent (SWE) estimates have been elusive,
particularly in mountain areas, however, there now appears to be some potential for direct satellite-based SWE observations along ground tracks that only cover a portion of a spatial domain (e.g., watershed). Fortunately, spatiotemporally continuous meteorological and surface variables could be leveraged to infer SWE in the gaps between satellite ground tracks. Here, we evaluate statistical and machine learning (ML) approaches to perform a track-to-area (TTA) transformation of synthetic SWE observations in California's
Upper Tuolumne River Watershed. We construct relationships between multiple meteorological and surface variables and synthetic SWE observations along observation tracks, and we then extend this relationship to unobserved areas between ground tracks to estimate SWE over the entire watershed. Domain-wide April 1st SWE inferred using two satellite tracks (~4.5% basin coverage) resulted in percent error of basin-averaged SWE of 24.5%, 4.5%, and 6.3% in an extreme dry year (WY2015), a normal year (WY2008) and an
extraordinarily wet year (WY2017), respectively. Assuming a 10-day overpass interval, percent errors in basin-averaged SWE in both snow accumulation and snowmelt seasons were mostly less than 10%. We employ feature sensitivity analysis to overcome the black-box nature of ML methods and increase the explainability of the ML results. Our feature sensitivity analysis shows that precipitation is the dominant variable controlling the TTA SWE estimation, followed by net longwave radiation. We find a modest increase
in SWE estimation accuracy when more than two ground tracks are leveraged. Accuracy of Apr 1st SWE estimation is only modestly improved for track repeats more often than about 15 days.



## 1 Introduction

Snow is a key component of the water cycle and a critical water resource for human and natural systems.
Seasonal snowpack serves as a natural "water tower" that stores water in winter and releases it during spring and early summer. It also shifts the time of peak runoff to be more aligned with the peak water demands from agricultural and municipal water users. It therefore, mitigates water shortages in summer and fall (Li et al., 2017). Snow-dominated river basins account for over half of the Northern Hemisphere land area, and seasonal snowpacks (and glaciers to a much lesser extent) provide water for over one-sixth of the world's
population (Barnett et al., 2005). Also, snow plays a crucial role in modulating the ecological functioning of terrestrial and aquatic ecosystems (Trujillo et al., 2012).

Snow Water Equivalent (SWE) is a measure of the amount of water stored in a snowpack; it is the depth of water that would result if the snowpack was melted. However, while in-situ measurements of SWE have long
been made at snow courses and more recently at automated snow pillows (which weigh snow accumulated on a measurement platform), these point-scale SWE measurements poorly characterize the spatial variability of SWE because of the relatively small number of observations and the significant under-sampling in high-elevation mountains where large amounts of snow accumulates (Dozier, 2011). In-situ observations are further complicated by the complex snow accumulation and ablation processes (Dong, 2018). In mountainous
areas, SWE has high spatial variability caused by complex physiographic and atmospheric conditions (Molotch and Bales, 2005, 2006), making SWE measurements even more challenging. Lettenmaier et al. (2015) state that spatial SWE data acquisition from satellite sensors has been elusive, especially in mountainous areas, and "deserves new strategic thinking from the hydrologic community".

Remote sensing is attractive for snow measurements over large areas because it avoids the need to access remote areas and complex terrain (Nolin, 2010; Guan et al., 2013; Schneider and Molotch, 2016). Remote sensing also has the potential to provide spatial, rather than point observations of SWE. Over the last forty years, many studies have examined the application of satellite-based passive microwave (PM) sensors for SWE retrieval. The interest in PM-based retrievals has been motivated by (1) over 40-years of daily or sub-
daily observation availability, and (2) the capability of making global SWE observations in cloudy conditions and darkness (except when during periods of precipitation) (Foster et al., 2005). However, a number of limitations of PM-based SWE observations such as coarse spatial resolution (tens of km), signal saturation for deep snow, and relatively large errors in forested and topographically complex areas etc. (Li et al., 2017), have severely restricted its use, especially in mountainous areas.

For these reasons, over the last few years, there has been a shift in interest in the mountain snow community to new technologies that have the potential to obtain snow measurements with higher accuracy and spatial resolution. For example, LiDAR (Light Detection And Ranging) is an active remote sensing tool with the ability to retrieve precise snow depth at high spatial resolutions (albeit mostly along relatively narrow swaths)



in complex terrain and forested regions (Lefsky et al., 2002; Painter et al., 2016). P-band Signals of Opportunity (P-SoOp; ~1m wavelength) is an emerging technology in SWE measurement, which has the capability of penetrating through dense vegetation and into the root zone, with a reflection coefficient phase that is able to simultaneously measure SWE (of dry snow; depth of wet snow is retrievable) and root zone soil moisture (Garrison et al., 2019; Yueh et al., 2021). Although both LiDAR and P-band signals of

opportunity have potential advantages for SWE retrieval, due to orbital constraints, both methods would provide track (or narrow swath) observations rather than continuous SWE maps. However, snow distribution and snowmelt runoff generation are spatiotemporally continuous processes. Hence, developing "track-to-area" (TTA) transformation would provide derived space-time continuous SWE that can significantly increase the utility and value of the remote sensing observations.


TTA transformation could be achieved by leveraging snow pattern repeatability. For instance, Pflug and Lundquist (2020) inferred the spatial distribution of snow depth on April 7$^{th}$, 2014 in California's Tuolumne watershed using snow depth observations subsampled across only a portion of the study domain (< 4%) and observed snow patterns from a different water year. Their results for inferred distributed snow depth had a

mean absolute error of less than about 10%.

P-SoOP has the potential to be deployed on space platforms to provide direct SWE and root zone soil moisture measurements during snow accumulation periods, including near the time of peak SWE that is the most significant time for water management (Shah et al., 2018). For example, NASA's SNoOPI satellite (SigNals

of Opportunity: P-band Investigation) has been planned for P-SoOp space demonstration (https://www.nasa.gov/feature/goddard/2019/snoopi-a-flying-ace-for-soil-moisture-and-snow-measurements). TTA transformation would be required to obtain domain-wide SWE estimation based on the track-based P-SoOp SWE observations. Potential tools include interpolation, statistical models, data assimilation, and machine learning (ML). In the case of ML, along-track SWE observations would be

leveraged as training samples for ML algorithms used to produce SWE estimates over the observation gaps between ground tracks.

Here, we utilize the Western U.S. Snow Reanalysis data (WUS-SR; Fang et al. 2022, hereafter F2022) as "truth" from which we synthesize P-band SWE observations along tracks, and in turn to explore TTA

transformation strategies. The TTA transformation of along-track SWE observations are achieved using statistical and ML approaches. Specifically, we address the following four questions: (1) How does the spatially distributed April 1$^{st}$ SWE inferred from TTA compare with the synthetic truth, and how do their differences vary in dry, normal, and wet years? (2) What are the dominant variables for the April 1$^{st}$ SWE estimation in statistical and machine learning TTA methods, and which method has the highest accuracy? (3)

How does the accuracy of the domain-wide SWE estimates from TTA approaches evolve within a season at different temporal observation resolution? (4) How does the performance of TTA change as a function of the



spatial sampling density (number of hypothetical ground tracks), and what is the preferred number of tracks? Our study is intended to provide a pathway forward in support of future snow satellite design and SWE estimation over mountainous areas globally.

## 2 Study area and data

### 2.1 Study area

Our study area is the Upper Tuolumne River Basin (above Hetch Hetchy Reservoir) in the Sierra Nevada of California. This Tuolumne basin has a drainage area of approximately 1650 km$^2$ that is characterized by complex high-elevation topography (Fig. 1 and Fig. S1). Elevations in the watershed range from about 700 to 3900 m, with most of the basin area located above 2500 m (Fig. 1). Slopes are distributed between 0° and greater than 50° and the terrain surface mostly has NW and SE facing aspects. Fractional vegetation cover ranges from 0% (in high-elevation areas) to up to 60% in low-elevation areas. The runoff in the Upper Tuolumne River Basin is snow-dominant with a substantial high elevation contribution (30% of its runoff originates from elevations of 3000 m and above). In this respect, it is typical of many river basins that head in the Sierra Nevada and supply much of California's water.

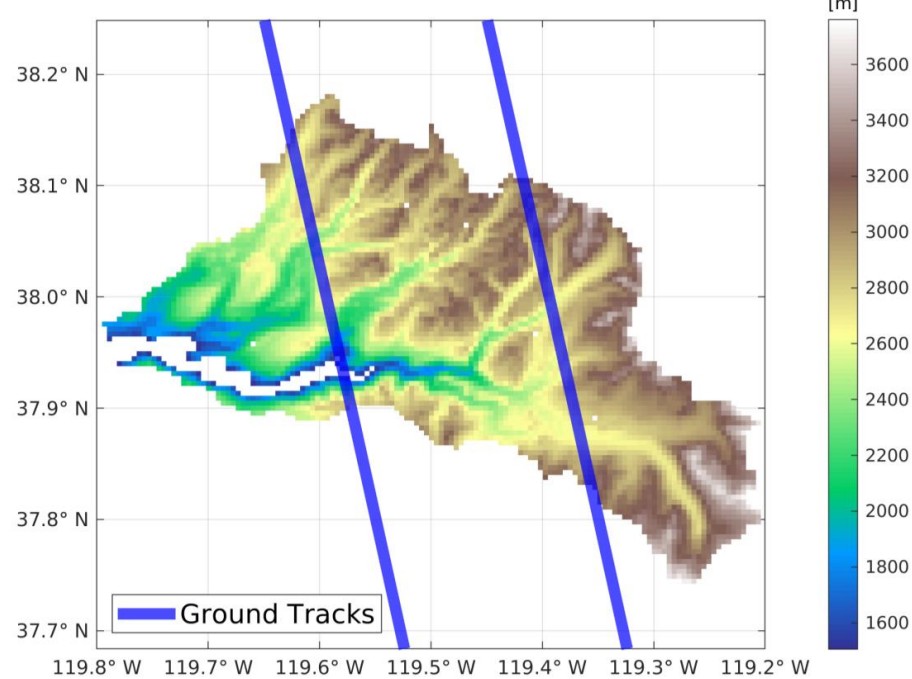

**Figure 1. Elevation of the Upper Tuolumne River Basin (above Hetch Hetchy Reservoir). Blue lines are the synthetic ground tracks passing through the study domain. The hypothetical tracks are about 1 km wide and the distance between the two tracks is approximately 21 km.**



## 2.2 data

We leveraged the F2022 snow reanalysis data as the basis for our synthesis of satellite observations along ground tracks. All the synthetic tracks are 960 meter wide (two 480 m reanalysis data pixels in width). This dataset was generated (by F2022) based on a Bayesian snow reanalysis framework with ensemble prior SWE

estimates updated by assimilating fractional snow-covered area (fSCA) observations from the Landsat satellite platforms using a Particle Batch Smoother approach (Margulis et al., 2015). Prior SWE estimates (required by the data assimilation approach) were derived from the land surface model SSiB-SAST (Sun and Xue, 2001; Xue et al., 2003) with the Liston (2004) snow depletion curve. F2022 shows that the reanalysis SWE estimates match in situ observations of peak SWE well across the Sierra Nevada, with mean difference

of -13 cm and correlation coefficient 0.86 taken over 1432 site-years of observations.

We sub-sampled the snow reanalysis data along the postulated ground tracks to synthesize the SWE observations that P-band sensors would produce (see section 3.1 for details). We also used F2022 as the SWE "truth" to evaluate our TTA SWE data transformation accuracy.


We used meteorological variables and static parameters including topographical characteristics and vegetation cover data as the training inputs (along ground tracks) for ML algorithms and as the model inputs (full-domain) for the domain-wide statistical and ML methods. Meteorological forcings included precipitation (PPT), air temperature (Ta), surface air pressure (Ps), specific humidity (q), net shortwave

radiation (NetShort), net longwave radiation (NetLong), and wind speed (wind). The meteorological forcing fields were obtained from the Modern-Era Retrospective analysis for Research and Applications, version 2 (MERRA-2; Gelaro et al., 2017) updated via the F2022 snow data assimilation. The digital elevation model (DEM) was obtained from the Shuttle Radar Topography Mission (SRTM; (Farr et al., 2007)), with gaps filled with the Advanced Spaceborne Thermal Emission and Reflection Radiometer (ASTER) Global Digital

Elevation Model (GDEM, version 2) product. The original spatial resolution of these two DEMs was 1 arc-second. The fractional vegetation cover data were taken from the Tree Canopy Cover (TCC) product containing the Landsat Vegetation Continuous Fields (Sexton et al., 2013). The meteorological, topographic, and land cover data were resampled to the same spatial resolution of the snow reanalysis dataset (i.e., 16 arc-second). In real applications, the meteorological forcings could come from any multi-source surface and

weather modeling data, e.g., weather forecast model analysis (real-time) or reanalysis (retrospective) fields.



## 3 Methodology

### 3.1 Experiment design

We addressed our four research questions (section 1) via four TTA experiments. Each of the experiments used four algorithms: one statistical and three ML methods (details in sections 3.2 and 3.3). For all the TTA experiments, the general idea is that we use the algorithms to build a connection between the observed SWE and the meteorological and static variables along the ground tracks on the observation days; these relationships reflect the physical control of the meteorological and static variables on SWE under different terrain, landscape, meteorological, and climatic conditions. We then extend these relationships to the unobserved areas and periods where and when meteorological and static variables are available to estimate SWE across the entire basin.

In the experiments, the target day can be any date (not necessarily one for which there is a satellite overpass). We used observations with close temporal proximity to train the four algorithms. For example, if we intend to fill the spatial SWE gaps on April $1^{st}$, our target day would be April $1^{st}$ and we used the SWE (track) observations available within a short period before or on the target day for model training.

We first focused on estimating spatially continuous peak annual SWE in the basin because of its water resource importance (section 3.2.1). We then explored the seasonal evolution of TTA SWE estimation, where we sequentially set the target day to be all days within a water year to obtain spatially and temporally continuous SWE estimates over the entire Tuolumne basin (section 3.2.2). Furthermore, we conducted two experiments to evaluate the impact of meteorological variable and SWE sample density on the accuracy of the SWE estimates (sections 3.2.3 and 3.2.4). The two experiments introduced in sections 3.2.3 and 3.2.4 aim to align with explainable AI (e.g. Chakraborty et al., 2021; Dikshit and Pradhan, 2021a, b; Kratzert et al., 2019), which facilitates the comprehension and trust of the results and outputs created by ML algorithms. A major objective of explainable AI is to overcome the black-box nature of ML systems, which is particularly important for hydrologic applications that are mostly process-oriented.

We used four metrics to quantify the performance of our TTA experiments: (1) mean absolute errors (MAE), (2) median ($50^{th}$ percentile) of percent absolute error at a pixel-level (PAE_50), (3) $90^{th}$ percentile of percent absolute error at a pixel level (PAE_90), and (4) percent error of basin-averaged SWE (PEBAS). When calculating PAE_50 and PAE_90, we first calculated the percent absolute error of SWE estimates for each pixel within the study area, and then found the median and the $90^{th}$ percentile of the pixel-level percent errors. To avoid extremely high percent errors due to zero or nearly-zero SWE values, we filtered out pixels with SWE truth less than 0.05 m when calculating PAE_50 and PAE_90. For annual peak SWE estimation (see section 3.2.1 and 4.1 for details), we also calculated the bias ratio to quantify the degree of over or underestimates of our TTA transformations.



### 3.2 Model training, estimation, and output correction

#### 3.2.1 Annual peak SWE estimation

In the annual peak SWE estimation experiment, we sought to fill spatial gaps between ground tracks on April 1st of three target years: WY2015, WY2008, and WY2017. April 1st SWE has long been used as a proxy of peak snow water resource availability and is a critical variable for seasonal streamflow forecasting. WY2015, 2008, and 2017 represented extremely dry, normal, and extremely wet years in the Upper Tuolumne basin. These three years had the lowest (~50.4% of the average of the MERRA2 grided-based precipitation between WY1985 to 2019), normal (~96.8% of average), and highest (~174.0% of average) winter (Nov 1st to Mar 31st) precipitation from WY1985 to 2019, respectively. To train the models for each of the three target years, we assumed that we had seven SWE observations before and on April 1st in the target year and the two years ahead of that target year, and the temporal interval between observations within each of the three years was five days. For example, for the SWE TTA on April 1st WY2008, we used observations from late February to April 1st of WY2008, WY2007, and WY2006. On each observation day, we assumed that there were two ground tracks at the same locations that cover approximately 4.5% of the study area (Fig. 1).

The training target is to reproduce the synthetic SWE observations along the two ground tracks. The training input features include the 5-day averaged meteorological forcings within each 5-day observation cycle (i.e. each observation day and the four days ahead) and static variables, which include topographical and vegetation cover data along the two hypothetical ground tracks (Fig. 1). The training builds the connection between all available training input features and target pairs. After the connections are built along the ground tracks (i.e. models are trained), we used the domain-wide 5-day averaged meteorological forcings (i.e. from March 28th to April 1st) and static variables as the input to the trained models to estimate domain-wide SWE on April 1st in the target year.

After the estimation step, we implemented an error correction to the domain-wide SWE estimates. Specifically, we first conducted probability density function (PDF) matching between the estimated SWE on the ground tracks and the synthetic true SWE along the ground tracks and applied the derived PDF correction to the off-track pixels. These corrections aimed to leverage the observations on the synthetic ground tracks to eliminate systematic biases and large errors in domain-wide SWE estimates.

#### 3.2.2 Seasonal basin-wide SWE estimation

We also applied the TTA transformation for each day over a full WY, assuming that the temporal interval between satellite observations was 0, 5, 10, 15, 20 or 30 days. We investigated the performance the SWE TTA estimation in different phases of a snow season (i.e. accumulation, peak, and melting periods) and the sensitivity of the performance to the observation frequency.





The seasonal SWE TTA transformation filled both the spatial and temporal gaps of SWE observations. In the seasonal TTA estimation with a fixed observation interval, on the days with SWE observations, we only need to fill the spatial gaps, so the training and estimation processes in this case are identical with the April 1st experiment (as in section 3.2.1). During the temporal gaps between SWE observations, the target days have no SWE observations to train the statistical and ML models, so we borrowed the established model from the

closest previous observation day and input the domain-wide forcings (5-day averaged before and on the target day) and static variables to the borrowed models from the that closest previous observation day to obtain domain-wide SWE estimates on this non-observed day. We performed a PDF matching correction for the domain-wide estimates based on the closest (in time) previous observations on ground tracks.

After SWE estimation was implemented for every day over a full water year, we obtained daily and spatially continuous SWE maps for the Upper Tuolumne Basin for full water years 2015, 2008, and 2017. We calculated the daily time series of basin-averaged SWE by averaging SWE values for all pixels in the study domain and compared the estimated daily basin-averaged SWE with that computed from the synthetic truth.

### 3.2.3 Sensitivity of TTA to input meteorological forcings

The analysis of the sensitivity of TTA to the input meteorological forcing fields was performed to the April 1st SWE TTA transformation, so the training and estimation setups were the same as those in section 3.2.1, except that we employed the following two methods to investigate the sensitivity of the basin-wide SWE estimates to the input meteorological forcings:

1. Missing feature analysis: we withheld one training meteorological variable during the training process each time and re-trained all the four models with the remaining training fields. The change in the estimated basin-wide SWE compared with the original SWE estimates (i.e. the outputs from the model trained with all the forcing fields) could reflect the influence of this missing feature on domain-wide SWE estimates. We normalized the absolute change of MAE as an indicator to quantify the relative contribution and the

magnitude of influences of each meteorological forcing field to SWE estimation.

2. Forcing uncertainty analysis: for each pixel, we perturbed each training meteorological field with a percentage error (-50% to 50% with an interval of 1%), and each time we perturb only one field while holding the other forcing fields unchanged. A 0% error meant that the meteorological inputs were the same as their

original values (i.e., the same as we used in the experiment described above). Every time we added more error to a training field, we re-trained the statistical and ML models. We then used the trained model to predict the basin-wide SWE and used MAE to quantify the SWE estimate errors caused by the error perturbation. With the 100 realizations for each training field (-50% error to 50% error with an interval of 1%), we explored the corresponding changes in domain-wide SWE estimates, which allowed us to determine the influence of

forcing errors on SWE estimates and identify sources of model errors.





### 3.2.4 Sensitivity to the number of ground tracks

The investigations discussed above were all based on the two hypothetical ground tracks shown in Fig. 1. To explore the relationship between the number of ground tracks and estimation accuracy, we assumed that there were 1-6 overpasses covering from 2.42% to 12.10% of the study area on April 1st, so that the available
observations for model training vary with the different numbers of tracks passing through the river basin. All the tracks in each scenario were distributed over the entire Tuolumne basin with equal spacing. The training and estimation processes were the same as the April 1st experiment (details in section 3.2.1).

### 3.3 Satellite observation gap-filling methods

We utilized and compared the four algorithms to transform the postulated track-based satellite observations into space-continuous SWE estimates, as described below.

### 3.3.1 Statistical method

As applied, multivariate linear regression (MVLR) defines a linear relationship between multiple independent
variables (input variables) and one dependent variable (the target variable) based on pre-defined rules, e.g., the regressed results are Best Linear Unbiased Estimates (BLUE) of the dependent variable (see Text S1 for details). In our case, the input variables were the meteorological forcings and static land cover features; the target variable was SWE.

### 3.3.2 Machine learning algorithms

We explored three machine learning (ML) methods: random forest (RF; Breiman, 2001), support vector machines (SVM; Vapnik, 1982), and deep neural networks (DNN; Tanaka and Okutomi, 2014) on building the relationship between inputs and SWE along ground tracks. The hyperparameters of the three ML methods were optimized using 10-fold cross validation. After the selection of model hyperparameters, we train each model for ten times. During each of the 10 training cycles, we randomly reserved 15% of the training dataset
as the test dataset that was used for evaluating the estimation results, and we trained the three ML models using the remaining 85% of the training dataset and estimated model performance using the test dataset. The 10-fold cross validation repeated this training-validating process 10 times with the training (85%) and validation (15%) sub dataset randomly selected each time. After the 10 cycles, we selected the five model setups with the lowest MAE for the test dataset and used these five selected model sets with domain-wide
input features to obtain five sets of SWE estimates over the whole watershed. Our final domain-wide SWE estimates were the average of the SWE estimates from the 5 selected models.



### 3.3.2.1 Random Forest

We used the random forest (RF) method introduced by Breiman (2001) implemented to simulate the non-linear relationship between input features and SWE. The basic building units of RF are an ensemble of
decision trees (DTs) that split a subset of features on each split (Kuter, 2021). Usually, a series of DTs is employed to achieve sufficient accuracy of final prediction by weighted averaging the prediction results of multiple selected DTs (Liu et al., 2020). The selection of decision trees was carried out by voting, that is, the higher the repetition degree of the DT, the higher the contribution of this DT to the RF model.

During the training process, we optimized two hyperparameters in the random forest system: (1) Ntree, the number of decision trees grown based on a bootstrap sample of observations; (2) Sleaf: minimum number of observations per tree leaf. One useful characteristic of RF is that it is a self-explainable model where the implementation and examination of the out-of-bag score is a form of model validation. To optimize the two hyperparameters, we carried out 10-fold cross-validation to find the optimal hyperparameter combinations
with the lowest out-of-bag errors. The change of errors with Ntree and Sleaf were shown in Fig. S2 and Fig. S3. Our analysis showed that the preferred number of decision trees was 50 and the minimum leaf size was 5.

### 3.3.2.2 Support vector regression

SVM is a supervised and non-parametric machine learning algorithm (Vapnik, 1982). For regression-based
SVM, the basic logic behind the learning task is to find a function that has the universal minimum deviation from the measured response values for the full range of observations (Vapnik, 1998).

During the training process of the SVM method, we mainly optimized two hyperparameters: (1) the kernel function, which specifies the method used to transform inputs to the required target, and (2) the kernel scale, which is a scaling parameter for the input data. Based on 10-fold cross validation, we specified the Gaussian
kernel function and selected the kernel scale based on a heuristic procedure, which used the subsampling and set a random number seed before training, so estimates can vary for every running process.

### 3.3.2.3 Deep Neural Network

Artificial neural network (ANN) builds a non-linear relationship between the independent variables and the
target variable by connecting neurons in one layer to the previous or next layers. In general, ANN is a multi-layer structure that includes an input layer, one hidden layer, and an output layer. The hidden layer consists of several neurons, each of which is assigned a weight. The output of each neuron is multiplied by the weight and serves as the input for a non-linear activation function (Abiodun et al., 2018). Usually, a simple ANN (i.e., with only one hidden layer) is capable of learning non-linear relationships between inputs and the target
(Rumelhart et al., 1986), while deep neural networks (DNN) with more than two hidden layers can learn more complicated relationships between inputs and outputs (Hinton et al., 2006). We tested several




combinations of the number of hidden layers and the number of neurons in each hidden layer and constructed a seven-layer neural network (which was essentially a Multilayer Perceptron (MLP)) with 10, 9, 8, 7, 6, 5, and 4 neurons in each layer, respectively. We chose Rectified Linear Units (ReLu) as the activation function in each hidden layer. The cost function used in this network is the Levenberg-Marquardt, which is considered as one of the most efficient learning algorithms in terms of convergence speed (Costa et al., 2007).

## 4. Results and discussion

### 4.1 Basin-wide April 1st SWE estimation

Figure 2 shows April 1st pixel-level results for all four algorithms for WY 2008, 2015, and 2017. DNN generally outperforms the other three methods, which has fewer outliers with results distributed closer to the 1:1 line on the scatterplot. Statistically, domain-wide SWE estimates from DNN are also the best among the four methods in terms of (1) lowest values of MAE (Fig. 2); (2) highest accuracy from the perspective of PEBAS (Fig. 3); and (3) at a pixel level, lowest values of PAE_50 and PAE_90 (Fig. 3).

Accurate SWE estimation in the extremely dry year is of key importance for water management in California. Figure 2 (d) shows that domain-wide SWE estimates in the extremely dry year (WY2015) are nearly unbiased for MVLR and DNN, but RF and SVM tend to overestimate SWE (Fig. 2). All ML-based domain-wide SWE estimates in WY2015 have higher accuracy than the statistical method (Fig. 2 and Fig. 3). DNN performs the best among the four algorithms in WY2015 in terms of MAE (0.028 m), PAE_50 (20.0%), PAE_90 (76.0%), and PEBAS (24.5%).

ML methods also are more accurate than the statistical method in the typical ("normal") year (WY2008). Compared to the three ML algorithms, the statistical method has the largest MAE, PAE_50, PAE_90, and PEBAS in WY2008 (Fig. 3). Compared to the dry year, SWE estimates are more accurate in the normal year in terms of PAE_50, PAE_90, and PEBAS for all the four algorithms. Possible reasons for the better performance in the normal year relative to the dry year are (1) the number of pixels with zero SWE value is much less in the normal year than in the dry year, so the useful training information for building the relationship between inputs and the target are more abundant in the normal year; (2) there are fewer pixels with small values of SWE in the normal year than in the dry year; small SWE values tend to generate large values in percentage error calculation, and so the values of metrics regarding percent errors are larger in the dry year. The reason for large values of PAE_90 in the dry year (DNN: 76.0%) than in the normal year (DNN: 38.4%) is that although we omitted pixels with extremely small SWE (< 0.05 m), in the dry year, there still are more pixels with low SWE, which are prone to high percent errors. The TTA SWE estimation is weakest in low-SWE situations, resulting in heavy tailed behaviour in the percent SWE errors under that condition.





ML-based estimates were also most accurate in the extremely wet year (WY2017). According to the estimation statistics, DNN is the best algorithm among the four TTA transformation methods with the lowest

MAE (0.22 m), PAE_50 (15.4%), PAE_90 (50.2%), and PEBAS (6.3%). Compared to the normal and dry years, the wet year has fewer pixels with zero or nearly-zero SWE values, thus the number of useful pixels for training ML algorithms is larger. Also, SWE values are larger, which tends to reduce the percent absolute errors, making PAE_50 and PAE_90 values generally smaller than those in the dry or normal years.

Overall, DNN outperforms the other three algorithms in all three years (Fig. 2 and Fig. 3), while the statistical method (MVLR) has larger values of MAE, PAE_50, PAE_90, and PEBAS than all the ML methods for all the three years. Due to the superior performance of DNN, the following results and discussion are based on DNN only; results for the other three methods are included in the supplemental material.

Cumulative winter precipitation is highly correlated with April 1$^{st}$ SWE, so SWE estimation errors tend to be small in dry years and large in wet years. Correspondingly, as shown in the spatial maps of SWE estimates and estimation errors (Fig. 4), in WY2015, for nearly all the pixels within the study area, the overall estimation errors are within the range ±0.2 m (PAE_50: 20.0% and PEBAS: 24.5%). The error range is larger in WY2008 than that in the dry year, which is about ±0.3 m (PAE_50: 9.4% and PEBAS: 4.5%) and larger

still (±0.5 m) in WY2017 (PAE_50: 15.43% and PEBAS: 6.31%).

The spatial maps of DNN-based domain-wide SWE estimation errors (Fig. 4 d-f) show that the patterns of error distribution are similar for the three years, that is, underestimates are more likely to appear in the low-elevation areas in the western watershed (elevation range: around 1500 to 2800 m) while overestimates

appear mainly in the high-elevation areas in the northern parts of the watershed (elevation range: approximately 2800 to 3800 m), especially in the normal and wet years (i.e., WY2008 and WY 2017). A possible explanation for this error pattern is that during the training process, ML models would leave out some outliers, some of which are probably the extreme values in low or high elevation areas, thus the estimates from the ML systems may tend to approach an average situation, that is, predict higher for low

values and lower for high values. Pixels in the low-elevation areas generally have low SWE, therefore overestimates tend to occur in these regions; in contrast, underestimation tend to occur more for pixels in high elevation areas.

We also evaluate errors in domain-wide April 1$^{st}$ SWE for a larger number (12) of years (4 driest, 4 normal,

and 4 wettest years from WY2000 to 2019) to better understand the impacts of climate conditions on the accuracy of domain-wide SWE estimation near the time of peak SWE time (Fig. 5). The metrics used to quantify the accuracy of SWE estimation include MAE, PAE_50, PAE_90, PEBAS, and bias ratio (slope of the regression line (intercept was forced to be 0) between estimation and truth).





Our results indicate that overall, MAE of April 1st SWE estimates becomes larger as precipitation increases

(MAE: wet years > normal years > dry years). For example, MAE in WY2017 is twice as large as the average

MAE of the other years (MAE in WY2017: 0.220 m; average MAE of the other years: 0.079 m). This is

likely because SWE is largely determined by the amount of winter precipitation in the given year. To better

compare the performance of DNN-based TTA transformation in different water years with climate

conditions, we further show the PAE_50, PAE_90, and PEBAS for each of the 12 years. According to

PAE_50, at a pixel level, half of the pixels have absolute percent errors smaller than 20% (except for

WY2001) even in the four driest years when extremely low SWE values may lead to large values of percent

absolute errors for many pixels in the study area. As noted above, the PAE_90 values are relatively large in

the dry years; on the other hand, the values of PAE_90 are smaller than or close to 50% in the normal and

wet years, indicating that 90% of the pixels in the study area have relatively small SWE estimation errors. In

addition, the values of PEBAS are less than 20% for all years except for WY2015 (which has zero April 1st

SWE in many locations that had not previously been snow-free during the instrumental record).

Bias ratio (quantified by the regression slope between the SWE estimate and truth with intercept forced to be

0) provides information about the degree of over- or under-estimation of domain-wide SWE estimates. The

bias ratio for the 12 years (Fig. 5) indicates that DNN provides an approximately unbiased estimate of

domain-wide SWE across all climate conditions with slopes of the zero-intercept regressions all within the

range 0.9-1.1. In the normal years, all slope values are close to 1.0 (WY2003: 1.01; WY2002: 1.01; WY2008:

1.01; WY2016: 0.99). SWE estimation modestly degrades under dry and wet conditions with slight

underestimation of SWE (with bias ratio around 0.93) in the two driest years (Fig. 5).






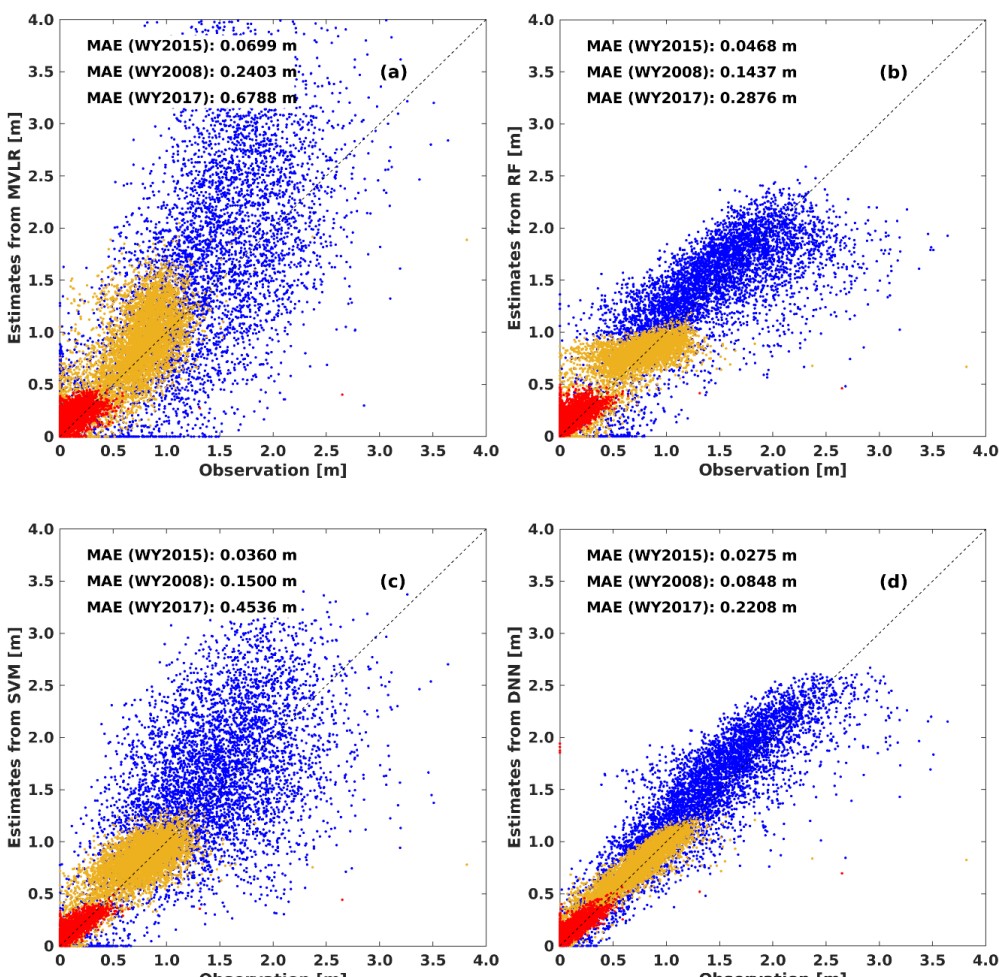

**Figure 2. Pixel-level scatterplots of April 1st SWE estimated by MVLR (a), RF (b), SVM (c), and DNN (d) versus the true April 1st SWE in WY2015 (dry year; red dots), WY2008 (normal year; yellow dots), and WY2017 (wet year; blue dots).**




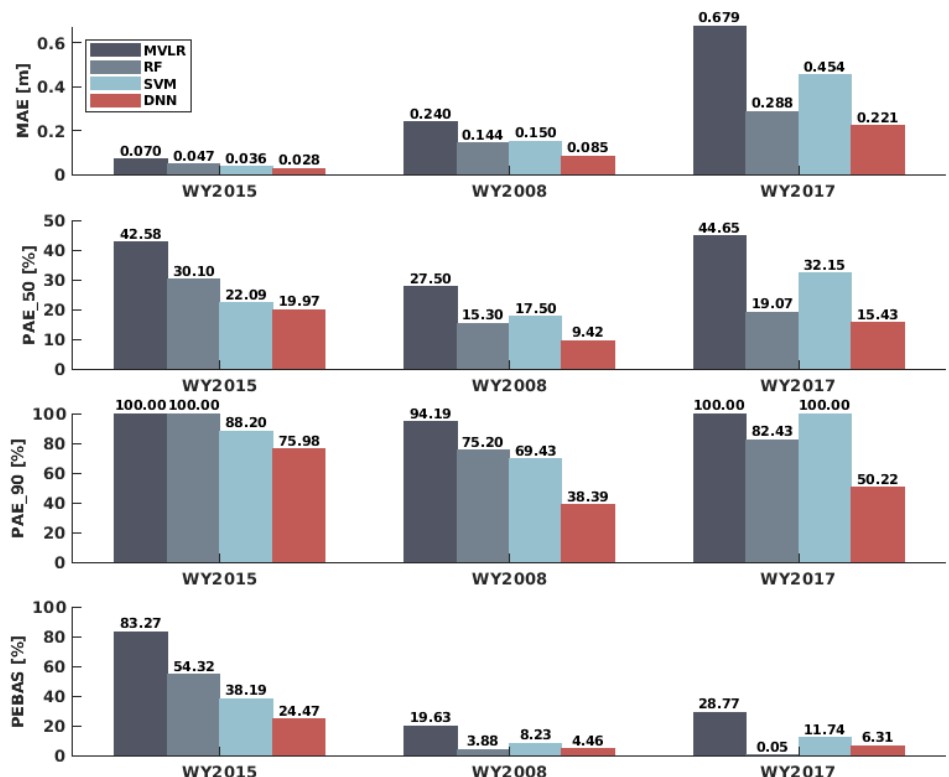

**Figure 3. Mean absolute error (MAE), median of percent absolute error (PAE_50), 90th percentile percent absolute error (PAE_90), and percent error of basin-averaged SWE (PEBAS) of domain-wide SWE estimates based on MVLR, RF, SVM, DNN in WY2015 (dry year), WY2008 (normal year), and WY2017 (wet year).**




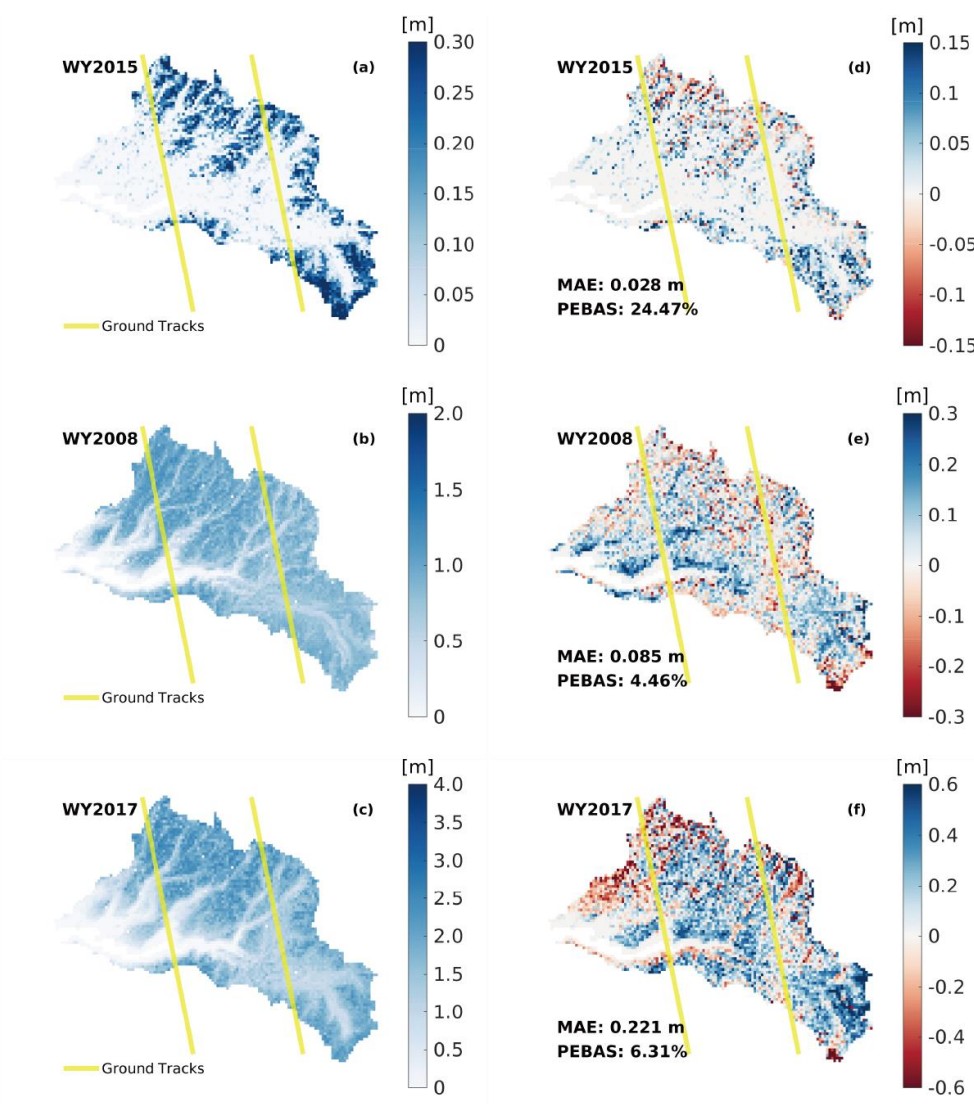

**Figure 4. DNN-Inferred April 1st SWE maps (a-c) and April 1st SWE errors (estimate minus truth; d-f) in WY2015, WY2008, and WY2017. Yellow lines are hypothetical ground tracks across the Upper Tuolumne Watershed, which are approximately 1 km wide and the distance between the two tracks is around 21 km.**




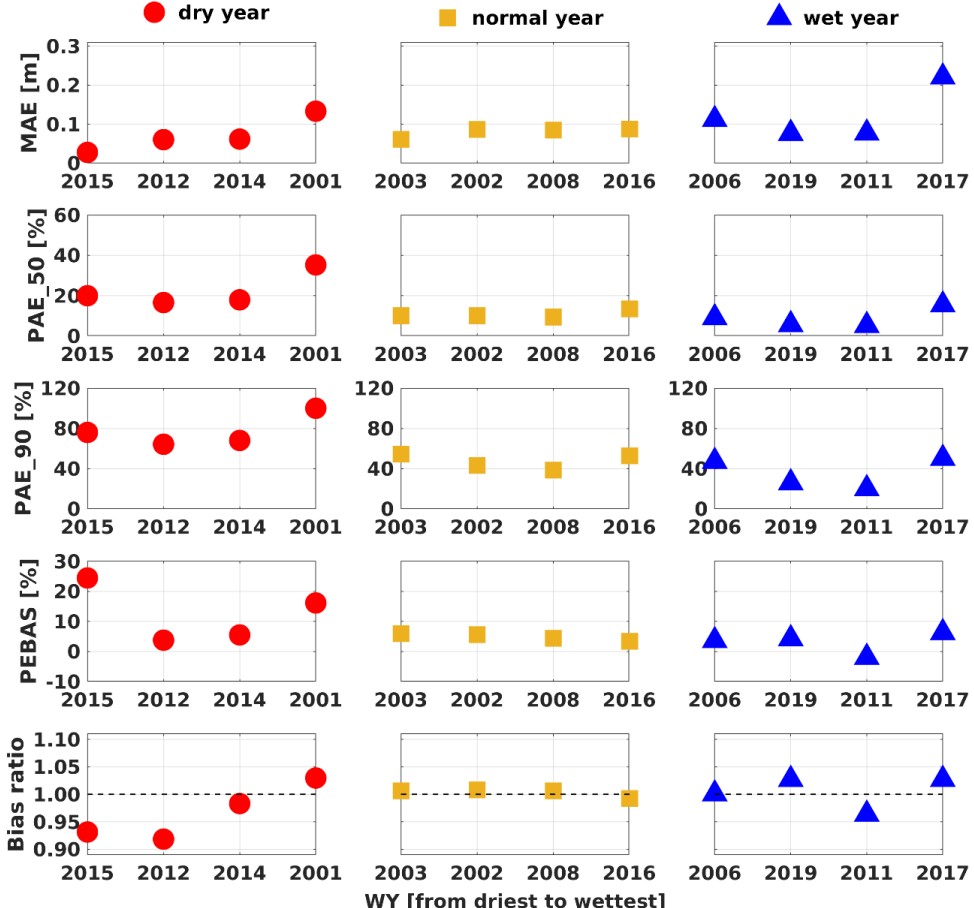

**Figure 5.** MAE (m; first row), PAE_50 (%; second row), PAE_90 (%; third row), PEBAS (%; fourth row), and bias ratio (slope; fifth row) of the DNN-based estimated April 1st SWE for the four driest years (red dots; WY2015, 2012, 2014, and 2001), four normal years (yellow square points; WY2003, 2002, 2008, and 2016), and four wettest years (blue triangle points; WY2006, 2009, 2011, and 2017) from WY2000 to WY2019.







**4.2 Daily time series of basin averaged SWE estimates**

Daily time series of basin-averaged SWE estimates based on DNN are shown in Fig. 6 for the three full water
years (dry average, and wet). The results of daily time series of basin-averaged SWE estimates based on DNN
for the three full water years (dry average, and wet) (Fig. 6) show that for satellite observations with daily
through 15-day revisits, the daily time series of SWE estimates is highly consistent with that of SWE truth
for all three years, aside from a slight overestimation of SWE around the time of peak SWE and during snow
ablation periods. The longer the interval between satellite overpasses, the larger the overestimation of
domain-wide SWE, especially for the days without snow observations. This is likely because the previous
TTA relationship applied to the unobserved dates is not well-suited for conditions on the target day, that is,
the delays between the TTA relationship and the domain-wide input features lead to overestimates near and
after the time of peak SWE. For 20-day and 30-day revisit intervals, this mismatch can be larger (up to 19 or
29 days), leading to large differences between SWE truth and SWE estimates (underestimation during snow
accumulation periods and overestimation during snow ablation seasons) are more obvious.

Daily time series of MAE during the snow accumulation season (January to April) and snowmelt season
(April to June) is shown in Fig. 7 (first column). Generally, MAE increases and has larger fluctuations as
the satellite revisit interval increases, especially in the extreme wet year (2017). In WY2017, the values of
MAE are mostly less than 0.3 m when observations are available daily and less than about 0.5 m up to 15-
day intervals. For revisit intervals greater than 20 days, the absolute averaged estimate errors exceed 0.8 m
for most of the snow accumulation and melt seasons.

The evolution of PAE_50, PAE_90, and PEBAS during the snow accumulation season and snowmelt season
(Fig. 7) shows that the errors increase with the time interval between overpasses. Differences in accuracy for
time intervals up to about 15 days are not apparent despite slight underestimation at the beginning of January
and overestimation near the end of May (especially for the normal year) become more apparent with longer
time intervals. For most days during snow accumulation and snowmelt periods, the values of PEBAS and
PAE_50 are less than 30%. Assuming a 10-day overpass interval, percent errors in basin-averaged SWE are
mostly less than 10%. However, as the overpass interval increases beyond 20 days, the values of PEBAS and
PAE_50 exceed 50% for most of the days from January to June. Also, the underestimates at the beginning of
the snow accumulation season and overestimates at the end of the snowmelt periods are much more apparent
for overpass intervals exceeding 20 days. From the perspective of PAE_90, except for the 1-day and 5-day
revisit scenarios, the values of PAE_90 are mostly larger than 50% from January to June. This is probably
because there are many low SWE pixels during snow accumulation and snowmelt seasons in addition to the
days near the time of peak SWE. With the longer time interval, the ability of SWE estimation degrades, so
that over 10% of the pixels (most of them are low-SWE pixels) in the study area have large percent absolute
errors despite different climate conditions.





Considering revisit intervals from 1 through 30 days, in general 5-day, 10-day, and 15-day intervals are
       plausible options that balance revisit frequency and estimation accuracy. The 1-day interval does not improve
       the results much relative to, for instance, 5-days, but performance for greater than 20-day revisits is
       substantially degraded.


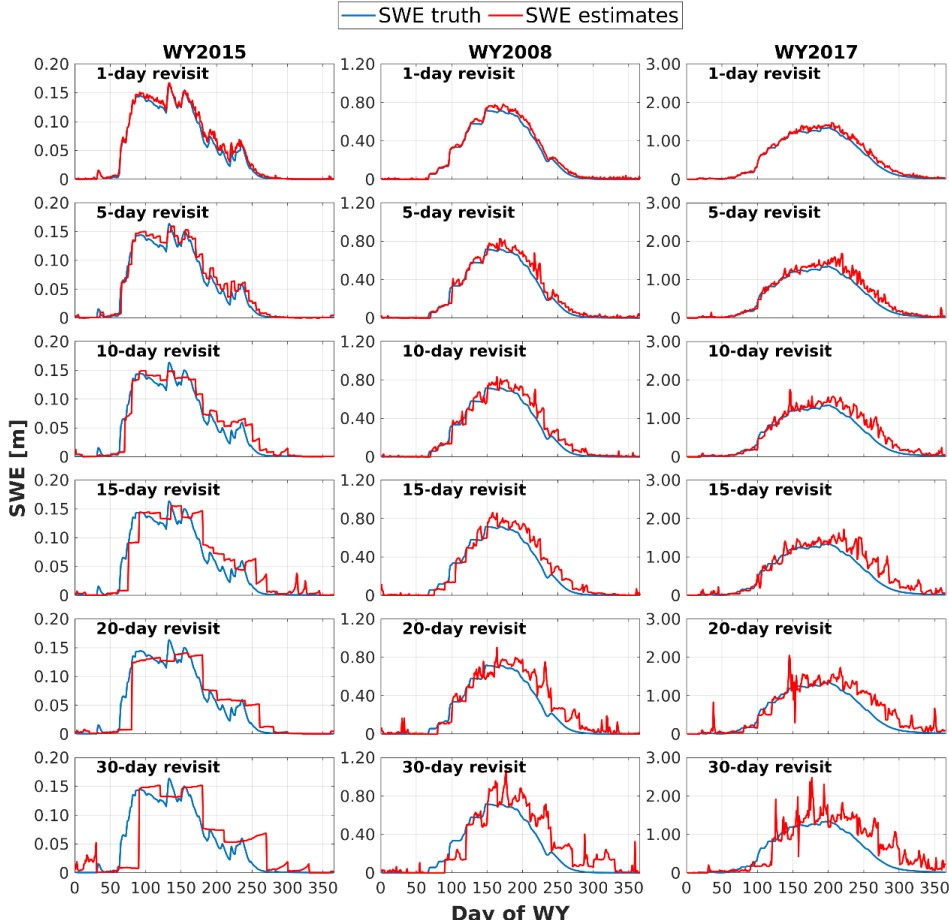

**Figure 6. Daily time series of basin-averaged SWE truth (blue line) and DNN-based SWE estimates (red line) in a dry year (WY2015), a normal year (WY2008), and a wet year (WY2017) for daily, 5-day, 10-day, 15-day, 20-day, and 30-day revisits (rows 1-6, respectively).**






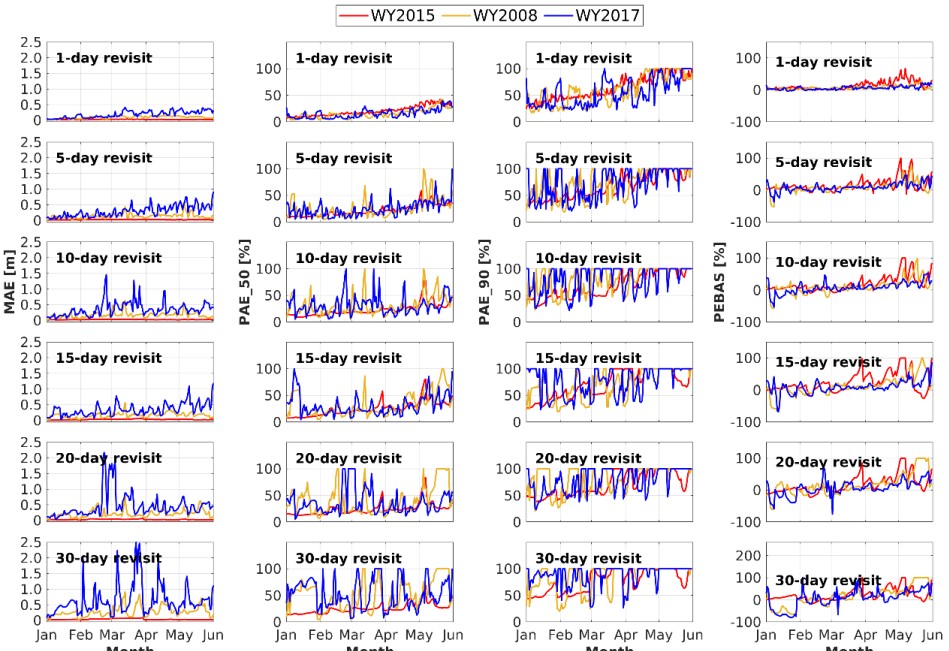

**Figure 7. Daily time series of MAE (m; first column), PAE_50 (%; second column), PAE_90 (%; third column), PEBAS (%; fourth column) from January to June in WY2015 (red line), 2008 (yellow line), and 2017 (blue line) based on revisit intervals of 1- through 30-days (rows 1-6, respectively).**


### 4.3 Input feature sensitivity test

The missing feature analysis evaluates the relative influence of each forcing field on the estimation of domain-wide SWE (Fig. 8). In the dry, normal, and wet years, precipitation is the dominant variable with relative contributions exceeding 50% (WY2015: 77.3%; WY2008: 50.6%; and WY2017: 50.3%), confirming that precipitation is the variable that provides the most the useful information for establishing the DNN-based TTA relationship regardless of climate conditions. The dominance of precipitation is the most significant in WY2015 with thinner snowpacks, the longevity of which is more sensitive to winter precipitation than in wetter years. In addition to precipitation, longwave radiation and shortwave radiation also play important roles in the domain-wide SWE estimation due to their critical controls on snowmelt rate and timing.

The changes of model performance as a result of the error perturbation in the training dataset (Fig. 9) show the potential influence of forcing biases on the SWE estimation results in the dry year (WY2015), normal year (WY2008), and wet year (WY2017). We explored the sensitivities of DNN-based (which is the best TTA transformation method) domain-wide SWE estimation results to different levels of biases perturbed to the training dataset. Figure 9 shows two points. First, the bias in the training meteorological features



propagate to the SWE estimates, especially in the years with normal and deep snow, which is not surprising because the bias affects the model training and larger bias have larger impacts. Second, the fluctuation in each MAE curve in Fig. 9 is obvious. The reasons for the fluctuation in the curves are that: (1) every time we add biases to the training meteorological data, we need to re-train the DNN-based TTA relationship. The weights assigned for each neuron in each hidden layers in DNNs have a degree of randomness, so even though the object of every DNN is to achieve the optimal estimation results, the inner structure of the DNNs are different to adapt to the biased training dataset; and (2) we only use 85% of training data that are randomly split from the original dataset (the remained 15% are used for model test). With different training data, each time we obtain a different DNN, so SWE estimates from the network are with slight differences.

In the extremely dry year (WY2015), the DNN-based domain-wide SWE estimates are not sensitive to the biases in the meteorological training inputs (Fig. 9), likely due to its extremely low snow cover. Among the seven meteorological forcings, Ta (primarily the positive biases of Ta) has relatively larger impacts on SWE estimate accuracy than the other meteorological forcings in WY2015. In the normal (WY2008) and wet (WY2017) years, positive biases of longwave radiation and negative biases of precipitation are the main sources of DNN-based SWE estimate errors (Fig. 9). In the years with relatively abundant precipitation (normal and wet years), precipitation is highly positively correlated with SWE values, so any precipitation errors in the training dataset can have a large impact on the accuracy of SWE estimation. In addition, We propose the following possible reason for the larger MAE caused by positive errors rather than negative errors: less than normal net longwave radiation can lead to delay or unmelted SWE near the original time of peak SWE, while larger net longwave radiation can cause earlier snowmelt and correspondingly, less SWE near April 1$^{st}$, so increased net longwave radiation would influence the SWE estimates more than decreases. In the DNN-based SWE estimates, errors in air pressure (Ps), air temperature (Ta), specific humidity (q), net shortwave radiation (NetShort), and wind speed (wind) have very small impacts on domain-wide SWE estimation under normal or wet climate conditions.

The robustness and stability of SWE estimate models are critical to estimating full-domain SWE in real applications. Overall, the performance of DNN degrades with more biases added to the training meteorological inputs in the normal and wet years, while the dry year is less sensitive to biases in the training data. Despite the fact that forcing biases can lead to lower SWE estimation accuracy in the normal and wet years, DNN-based SWE estimation has MAE < 0.3 m when the biases in training forcings are as large as ±50%, indicating the robustness of DNN in the TTA SWE transformation. The feature sensitivity results for the other three methods (MVLR, RF, and SVM) in the dry, normal, and wet years are shown in Fig. S4-6.

550




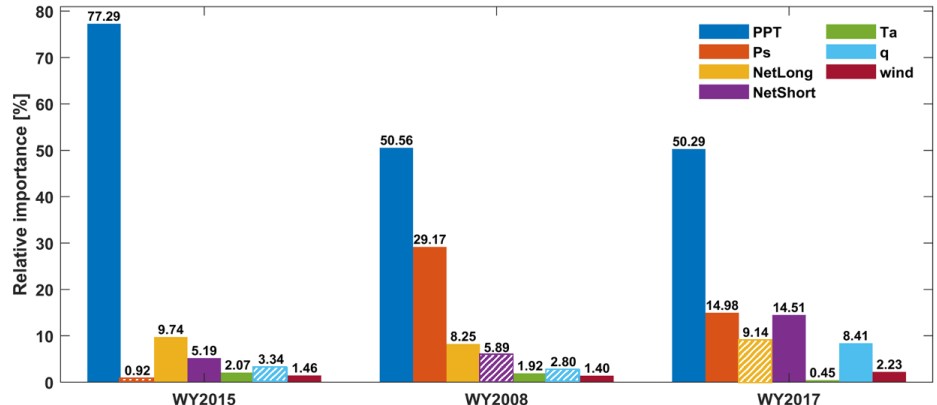

**Figure 8. Relative importance (%; normalized values of absolute change of MAE after removing one forcing field) of each meteorological forcing in DNN-based full-domain SWE estimates in WY2015, WY2008, and WY2017. The bars with dashed lines indicate that removing those variables decreases the value of MAE, while solid bars indicate that removing those variables increases MAE.**



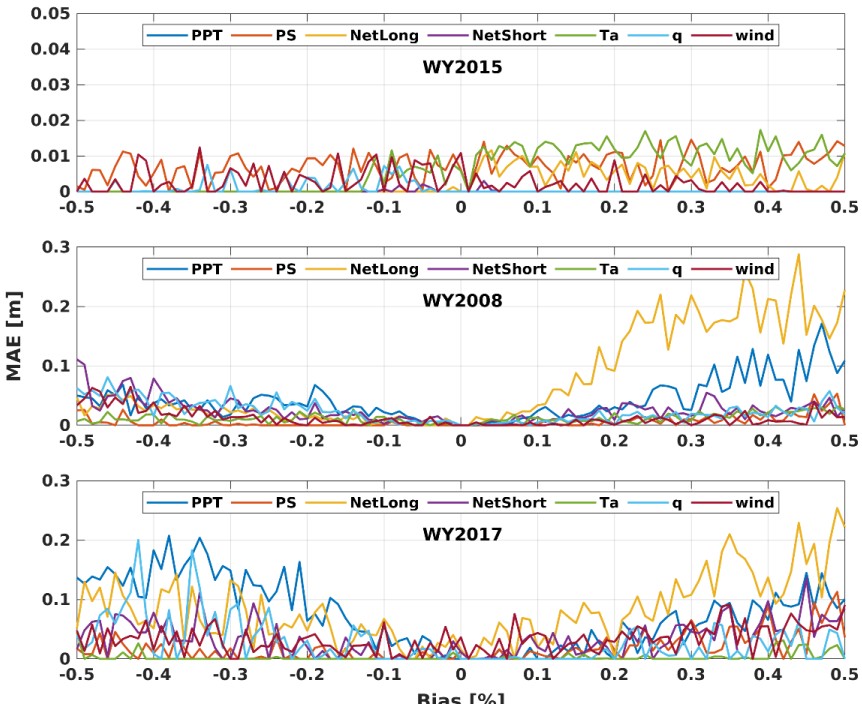

**Figure 9. Changes of MAE (m; relative to no-bias) of the inferred April 1st SWE in WY2015 (dry year; first row), WY2008 (normal year; second row), and WY2017 (wet year; third row) with biases perturbed in the meteorological forcings of the training datasets. The limit of the y-axis scale in the WY2015 panel is smaller than that of WY2008 and WY2017 to make the small MAE in WY2015 discernible.**

**4.4 Sensitivity of TTA to the number of ground tracks**

Figures 10 and 11 show the changes in MAE of the domain-wide April 1st SWE estimates with different numbers of ground tracks. We compared the SWE estimation errors in the dry, normal, and wet years (i.e., WY2015, 2008, 2017, respectively) using the DNN-based TTA method. In general, the performance of the DNN-based domain-wide SWE estimates improves with more ground tracks in the three years (Fig. 11). The improvement in the domain-wide SWE estimation is most distinct in the wet year, probably because more information for building the TTA relationship is available when snow accumulation is larger, and the number of pixels with zero or nearly-zero SWE is smaller.

Statistically, in WY2015, DNN estimates the basin-wide April 1st SWE with MAE less than 0.04 m when two or more ground tracks are available. Similarly, in WY2008 (a normal year), the DNN method has MAE





less than 0.10 m with two or more ground tracks. In all years, improvements in accuracy are small when the number of tracks exceeds two.

580  Regardless of snow climatologic conditions, the improvements of TTA performance with additional overpasses are limited when the number of ground tracks is larger than or equals to two. Based on the elevation distribution of pixels on synthetical ground tracks (Fig. S7), if more than two ground tracks passing through the study area, the useful information added to the training data become limited since pixels at different elevation bands seem to be similarly distributed, thus the decrease of MAE is limited. Also, the decrease of MAE as the number of the ground tracks increases from one to two could likely benefit from the
585  addition of training data in low-elevation regions (elevation < 2500 m). Considering the trade-off between SWE estimation accuracy and the cost of additional overpasses, one or two ground tracks is likely the optimal choice for purposes of domain-wide SWE estimation.

DNN-based SWE estimation is more accurate than the other methods regardless of the size of the training
590  dataset and the climate conditions. In contrast, the statistical method is the worst of the four (Fig. S8).




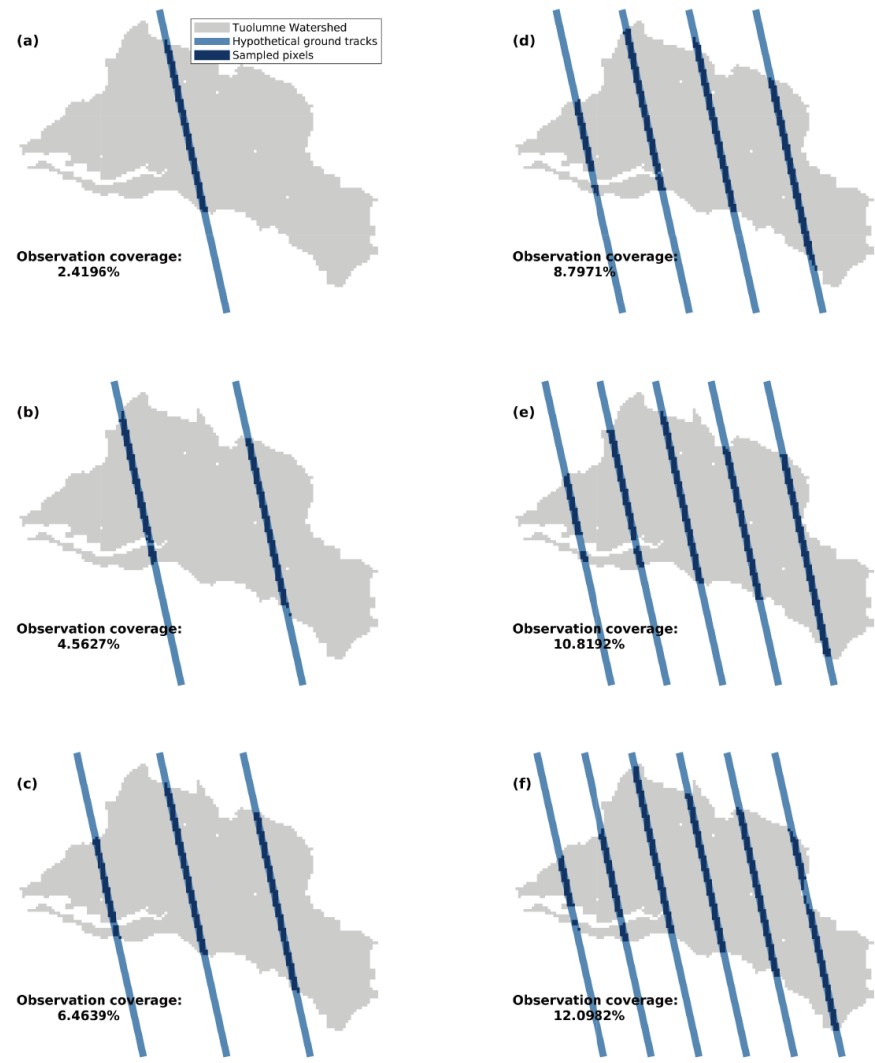

**Figure 10. Illustration of the 1-6 (a-f, respectively) hypothetical ground tracks in the Upper Tuolumne River Basin. The distance between each track is roughly the same over the whole watershed.**

595



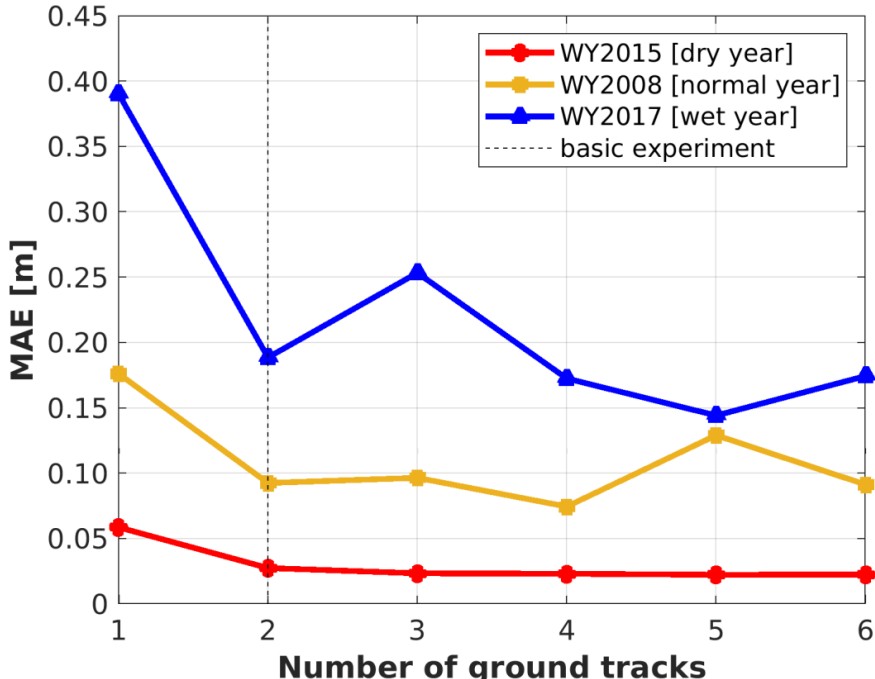

**Figure 11. Changes in MAE (m) of the DNN-based inferred April 1st SWE in dry (WY2015; red dots), normal (WY2008; yellow square points), and wet years with the number of ground tracks. The dashed lines for the number of ground tracks equal to 2 indicate that the addition of more satellite overpasses does improve the estimates much (see also section 4.1).**



**Conclusions**

Spatially continuous SWE estimates are of key importance to the prediction of the timing and volume of streamflow in snow-dominated regions. The potential now exists via at least two satellite-based technologies to measure SWE along tracks, which would cover only a small portion of a watershed's area. Fortunately, though, there exist relationships among multiple accessible variables (static and forcing fields) and SWE that can be use in linear or non-linear relationships to fill the gaps between tracks. Here, we use statistical and machine learning methods trained using the static variables, meteorological forcings, and SWE observations along tracks to estimate SWE over the entire domain (watershed). We tested (1) how the relationship between the training inputs and SWE along ground tracks could be used to infer SWE across the entire basin, (2) the performance of four algorithms applied over a full water year, (3) the influence of biases in meteorological forcings of the training dataset on the accuracy of the MVLR and three ML methods, and (4) changes in model performance with various numbers of overpasses. We focused on estimate accuracy over the Upper Tuolumne River Basin during dry (WY2015), normal (WY2008), and wet year (WY2017). Based on our results, we conclude that:

1. It is possible to derive basin-wide peak SWE (about April 1$^{st}$) with high accuracy (on the basis of MAE, PAE_50, PAE_90, and PEBAS) when the interval between satellite revisits is in the 5-10 days range.

2. The DNN method is most accurate of the four we tested regardless of snow climatological conditions. DNN is also the most robust method with respect to biases in forcing data and the reduction in the training data size. Though the DNN employed here is a simple MLP, it outperforms the statistical and the other two ML methods. It is reasonable to expect further improved performance of DNN with better network structure and hyper parameter optimization in future applications of snow data retrieval.

3. Based on missing feature analysis, precipitation is the dominant variable in domain-wide SWE estimation, especially in dry years. According to the results of our feature uncertainty analysis, the biases of precipitation and the net longwave radiation have the greatest influence on the accuracy of domain-wide SWE estimation.

4. As the number of ground tracks crossing the domain increases, the MAE of the inferred April 1$^{st}$ SWE improves, but only modestly when the number of ground tracks is more than two.

Our work demonstrates the feasibility of using ML algorithms (which almost always were more accurate than MVLR) to achieve TTA SWE estimates. Operationally, our feature sensitivity experiment provides a basis for determining the focus of quality control of meteorological forcings and the corresponding selection of TTA transformation methods. Furthermore, our exploration of the effects of addition of overpasses suggests the preferred balance between estimation accuracy and the number of satellite tracks: for the most part, increases in estimation accuracy are modest for more than two tracks. Further research could consider the

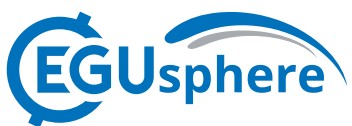

improvements of ML algorithms to improve the stability, efficiency, and accuracy of the TTA transformation systems. Lastly, due to the availabilities of the training datasets and accurate short-term forecast of

meteorological conditions (~1-2 weeks ahead), our ML methods can be used beyond the TTA framework for a history-to-future (HTF) snow estimation, where a trained relationship between historical snow and forcing fields across an area can be used in conjunction with the short-term meteorological forecasts to accurately forecast the SWE condition over the domain.

**Author contributions.** X.M. carried out the experiments and wrote the accompanying text. X.M., D.L., and D.P.L. designed the track to area methods. Y.F. and S.A.M. derived the snow reanalysis data and provided data description. All authors contributed to the project preparation, analyses, and manuscript writing.

**Data availability.** The data, code, and materials that can fully reproduce and extend the analyses in this paper

are archived in a public repository (https://doi.org/10.6084/m9.figshare.20044424.v1).

**Competing interests.** The authors declare no competing financial interest.

**Acknowledgement.** Xiaoyu Ma obtained stipend support from the China Scholarship Council (CSC) for 3

years during the doctoral study at the University of California, Los Angeles.






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
