# Peer review of "Estimating spatiotemporally continuous Snow Water Equivalent from intermittent satellite observations: An evaluation using synthetic data"

_EGUsphere, 2022_

## Author Comment (AC1)

**Response to Reviewers**

**Re: HESS Manuscript egusphere-2022-470: "Estimating spatiotemporally continuous snow water equivalent from intermittent satellite track observations using machine learning methods" by Ma et al.**

The comments from the editor and the reviewers are reproduced below in black and our corresponding responses are in blue.

Thanks for the great points and comments made by the two reviewers, which are entirely constructive and will, we believe improve the manuscript. We provide a point-by-point response below.

Reviewer #1 (Formal Review for Authors (shown to authors)):

Ma et al. investigate the use of machine learning for track-to-area transformation of satellite observations of SWE in swathes that cover a small fraction of a watershed (it is worth noting that point-to-area transformation with horizontal and vertical distance weighting has long been used for assimilation of in situ snow observations in numerical weather prediction). This study is entirely synthetic, but interesting in anticipation of real remote sensing opportunities. Some clarifications and numerous minor corrections are required. Explainability of the ML results is promised, but there is very little physical explanation in the discussion of the input feature sensitivity tests.

Thanks for the general comments on this manuscript. We will add more physical explanation and related references relative to the input feature sensitivity test as follows:

1.  Missing feature analysis:
    Winter precipitation in general is more influential than other meteorological forcings in SWE estimation (see e.g. Raleigh and Lundquist, 2012, Luce et al., 2014).

2.  Feature sensitivity analysis:
    Positive biases of net longwave radiation and biases in precipitation cause large SWE estimation errors in the normal and wet years (Sicart et al., 2006). This is because decreasing longwave radiation cannot increase 1 April SWE above the accumulated snowfall but increasing longwave radiation can decrease SWE all the way to zero (in theory), so increased net longwave radiation influences the SWE estimates more than decreases.

LiDAR can produce continuous SWE maps; the Airborne Snow Observatory does for this very basin.
We suggest deleting "LiDAR" here.  More generally, our intent was to demonstrate that the "track-to-area" method is useful for obtaining spatially continuous SWE where SWE observations are only available along ground tracks. In fact, we selected this basin because LiDAR data are available, but the idea is to demonstrate the track to area method in anticipation of a future satellite mission that provides data only along tracks, rather than images as ASO does.

"TTA transformation could be achieved by leveraging snow pattern repeatability", but that is not what is done here.

Here we aimed to introduce the application of other TTA or point-based SWE transformation methods. In addition to the method using snow pattern repeatability (Pflug and Lundquist, 2020), we intend, in the revised MS, to add and discuss briefly the following references:

(Magnusson et al., 2014)
Magnusson, J., Gustafsson, D., Hüsler, F., and Jonas, T.: Assimilation of point SWE data into a distributed snow cover model comparing two contrasting methods, Water Resources Research, 50, 7816–7835, https://doi.org/10.1002/2014WR015302, 2014.

(Schneider and Molotch, 2016)
Schneider, D. and Molotch, N. P.: Real-time estimation of snow water equivalent in the Upper Colorado River Basin using MODIS-based SWE Reconstructions and SNOTEL data, Water Resources Research, 52, 7892–7910, https://doi.org/10.1002/2016WR019067, 2016.

Is a 50% perturbation in temperature applied to the Kelvin temperature (which would be enormous) or Celsius temperature (which would be meaningless)? A 50% error in air pressure is also unphysical. To be clear, is the perturbation a fixed bias (as described) and not a random error? If so, why does the DNN not just learn the bias?
The unit here should be Celsius. The 50% error doesn't mean we increase or decrease 50% of the original value. Instead, we added 50% of the difference of maximum and minimum temperature for a specific pixel during the study period to the original value. The perturbation is a fixed bias. The reason why we added biases rather than random errors to meteorological inputs is that in snow hydrology, systematic SWE estimation biases is the primary source of estimation error, and it mainly comes from the biases of input meteorological forcings in SWE modeling. We intend to clarify this in the revised MS.

What is intended by saying "a simple ANN … is capable of learning non-linear relationships… while DNN … can learn more complicated relationships"? More complicated than nonlinear?
Our intention here was to point out the superiority of Multilayer Perceptron (MLP) relative to a single-layer Perceptron. A single-layer Perceptron is a neural network with only one neuron and can only understand linear relationships between the input and output data, while with Multilayer Perceptron, horizons are expanded and the neural network can have multiple layers of neurons, which are better adapted to more complex patterns (Gardner and Dorling, 1998). We will add clarification to this effect.

Levenberg-Marquardt is not the cost function, it is the algorithm used to minimize the cost function.
Thanks for pointing out the error. We suggest changing "cost function" to the "algorithm used to minimize the cost function".

Time series of SWE observations would not be available for wet snow after peak SWE.
We leverage the SWE reanalysis dataset (F2022 introduced in Section 2.2) as the synthetic "truth" for evaluating our SWE estimates. It is available over a full water year, whether or not the snow is wet. We will clarify this.

There is no discussion of why removing a particular variable will sometimes increase and sometimes decrease MAE in Figure 8.

We will add some discussion as to why some variables are not important in SWE estimation, and, for others that are, we will discuss their different roles in SWE estimation in a dry, normal, and wet year.

The biases in Figure 9 are fractions, not %. Why are there non-zero changes of MAE for zero bias in WY 2015 and 2017?

Thanks for pointing this out; we will change the X-axis by a factor of 100. The non-zero changes of MAE for zero bias were an error in plotting Figure 9, which is corrected below.

[Figure]

Figure 9.

I do not follow the argument for why positive longwave radiation errors cause larger errors. Is it that decreasing longwave radiation cannot increase 1 April SWE above the cumulated snowfall, but increasing longwave radiation can decrease SWE all the way to zero?

We will modify the explanation to make it more clear in the revised MS. Essentially decreasing longwave radiation cannot increase 1 April SWE above the accumulated snowfall but increasing longwave radiation can decrease SWE all the way to zero, so increased net longwave radiation influences the SWE estimates more than decreases.

The legend "Observation coverage" could be removed from Figure 10 so that it does not overlap the ground tracks (and four decimal places is excessive for the percentage).
Thanks for the suggestion, which we will follow. We will also move the information about observation coverage to the supplement (Table S1).

**Minor Corrections**
We've carefully proofread the MS and will make minor corrections throughout to eliminate the grammatical mistakes noted by the reviewer as indicated below.

"it therefore, mitigates" – delete comma
Revised.
"it therefore mitigates"

"whether large amounts of snow accumulate"
Revised.
"high-elevation mountains where large amounts of snow accumulates"

"spatial, rather than point observations" – delete comma
Revised.
"spatial rather than point observations"

"except during periods of precipitation"
Revised.

"topographically complex areas etc. (Le et al. 2017),"– delete etc. and comma
Revised.
"topographically complex areas (Le et al. 2017)"

"snow dominated"
Revised.

"Each of the experiments used one of four algorithms"
Revised.

"We investigated the performance of the SWE TTA estimation"
Revised.

"from the closest previous observation day"

Revised.

"for all four algorithms"

Revised.

"The reason for larger values"

Revised.

"for all three years"

Revised.

It would make more sense to say that April 1st SWE is highly correlated with cumulative winter precipitation.

Agreed. We will change this sentence to "April 1st SWE is highly correlated with cumulative winter precipitation."

"The error range is larger in WY2008 than in the dry year"

Revised.

"underestimation tends to occur"

Revised.

The information in this first sentence is repeated in the next sentence.

Revised. We will delete the redundant sentence.

"(dry, average and wet)"

Revised.

Delete "are more obvious"

Revised.

"are shown in Fig. 7"

Revised.

"becoming more apparent"

Revised.

"that provides the most useful information"
Revised.

"The dominance of precipitation is most significant"
Revised.

"larger biases have larger impacts"
Revised.

"so SWE estimates from the network are slightly different"
Revised.

"In addition, we propose"
Revised.

"is larger than two"
Revised.

"if more than two ground tracks pass"
Revised.

"more satellite overpasses do not improve the estimates much"
Revised.

"can be used"
Revised.

"and wet years"
Revised.

"The DNN method is the most accurate"
Revised.

"and reduction in the training data size"
Revised.

**References**

Luce, C. H., Lopez-Burgos, V., and Holden, Z.: Sensitivity of snowpack storage to precipitation and temperature using spatial and temporal analog models, Water Resour. Res., 50, 9447–9462, https://doi.org/10.1002/2013WR014844, 2014.

Raleigh, M. S. and Lundquist, J. D.: Comparing and combining SWE estimates from the SNOW-17 model using PRISM and SWE reconstruction, Water Resour. Res., 48, https://doi.org/10.1029/2011WR010542, 2012.

---

## Author Comment (AC2)

**Response to Reviewers**

**Re: HESS Manuscript egusphere-2022-470: "Estimating spatiotemporally continuous snow water equivalent from intermittent satellite track observations using machine learning methods" by Ma et al.**

The comments from the editor and the reviewers are reproduced below in black and our corresponding responses are in blue.

Thanks for the great points and comments made by the two reviewers, which are entirely constructive and will, we believe improve the manuscript. We provide a point-by-point response below.

Reviewer #2 (Formal Review for Authors (shown to authors)):

The manuscript 'Estimating spatiotemporally continuous snow water equivalent from intermittent satellite track observations using machine learning methods' by Ma et al. provides an interesting synthetic study regarding SWE observations from potential satellite tracks for track-to-area transformations. In general, I think such studies are important to be prepared for novel satellite missions and their 'real world' applications. The study is carried out in the Upper Toulomne River Basin in California in the Western United States. The authors apply statistic and machine learning techniques to answer the following four research questions:

(1) How does the spatially distributed April 1st SWE inferred from TTA compare with the synthetic truth, and how do their differences vary in dry, normal, and wet years?

(2) What are the dominant variables for the April 1st SWE estimation in statistical and machine learning TTA methods, and which method has the highest accuracy?

(3) How does the accuracy of the domain-wide SWE estimates from TTA approaches evolve within a season at different temporal observation resolution?

(4) How does the performance of TTA change as a function of the spatial sampling density (number of hypothetical ground tracks), and what is the preferred number of tracks?

The manuscript is well written and organized. I agree with all points raised by reviewer 1. In addition, I have some further points, which need clarification and/or additional information before publication.

Please change the unit of SWE in mm (instead of m or cm) throughout the paper (text, figures, tables), as mm is the unit for SWE.
We will change the unit of SWE from m to mm throughout the paper as suggested, including text, figures, and tables.

In general, please discuss if the pixel size and satellite track width of approx. 1 km is suitable to accurately describe the quantity of snow and SWE accumulation and ablation within the catchment (and, also for which applications is this resolution sufficient?). Could there be any limitations and need for higher spatial resolutions?
Thank you for the suggestions. The pixel size is 480m and the track width is about 1 km. We did not design the parameters, but just used the best available data. The synthetic tracks are consistent with a notional design for a P-band signal of opportunity proposed mission (by NASA's Jet

Propulsion Laboratory; see Yueh et al., 2021 for more information). The "truth" for the analysis of SWE estimates is the snow reanalysis data (F2022), which is at a 480 m spatial resolution. Designing a satellite with finer spatial resolution is expected to increase the training size of ML-based TTA SWE transformation, which may lead to better performance of continuous SWE estimation, but about 1-km seems to be what is now technically feasible. Due to space limitations, we only explored one SWE spatial resolution, and we prefer to defer analysis of the sensitivity of ML performance to observation resolution to future studies. We will add some discussion to this effect in the conclusions section.

Title: The title is a bit misleading. The study is fully synthetic, but the title 'promises' somehow 'real' satellite track observations. I would recommend to at least include the word 'synthetic' (or 'hypothetic') in the title. Moreover, as you use statistic and machine learning methods it would make sense to add also statistic in the title. Suggestion:
'Estimating spatiotemporally continuous snow water equivalent from intermittent synthetic satellite track observations using statistic and machine learning methods'.
Thank you for your suggestion. We propose to add "potential" to the title to avoid implying that the observations are real. We found that the other qualifiers make the title too cumbersome, and instead we suggest "The potential for estimating spatiotemporally continuous snow water equivalent from intermittent satellite track observations".

Abstract: The abstract should be revised: Please add information on the potential future satellite mission (P-SoOP), on which you relate your work on TTA, and give information on the spatial resolution of your synthetic satellite tracks.
Thank you for your suggestions. What is a bit tricky is that some of the technical background is in a NASA proposal under consideration, and we're limited in what we can say, On the other hand, the Yueh et al. (2021) paper, which we cite, can be leveraged for more information, including some on P-SoOP, which we will mention briefly in the abstract as the starting point for our synthetic SWE observations. We also will state in the abstract that that the spatial resolution of the synthetic satellite observations is 480 m.

38+: I am missing a solid literature review
   a)   on satellite based SWE and snow height derivation. This should at least include the
        following references:

   Lievens, H., Brangers, I., Marshall, H.P., Jonas, T., Olefs, M. and De Lannoy, G., 2022.
   Sentinel-1 snow depth retrieval at sub-kilometer resolution over the European Alps. The
   Cryosphere, 16(1), pp.159-177.

   Deschamps-Berger, C., Gascoin, S., Berthier, E., Deems, J., Gutmann, E., Dehecq, A., Shean,
   D. and Dumont, M., 2020. Snow depth mapping from stereo satellite imagery in mountainous
   terrain: evaluation using airborne laser-scanning data. The Cryosphere, 14(9), pp.2925-2940.

   b)   on other TTA or point-based methods.

   a)   Thank you for the suggestions, which we will add to the introduction.
   b)   In addition to the TTA and point-based method using snow pattern repeatability (Pflug
        and Lundquist, 2020), we will add the following references that discuss other TTA or
        point-based methods:

(Magnusson et al., 2014)
Magnusson, J., Gustafsson, D., Hüsler, F., and Jonas, T.: Assimilation of point SWE data into a distributed snow cover model comparing two contrasting methods, Water Resources Research, 50, 7816–7835, https://doi.org/10.1002/2014WR015302, 2014.

(Schneider and Molotch, 2016)
Schneider, D. and Molotch, N. P.: Real-time estimation of snow water equivalent in the Upper Colorado River Basin using MODIS-based SWE Reconstructions and SNO[TEL] data, Water Resources Research, 52, 7892–7910, https://doi.org/10.1002/2016WR019067, 2016.

57: Please add 'wet snow' as further limitation.
We will add a comment on this limitation of PM-based SWE remote sensing, with a reference to (Walker and Goodison, 1993).

82: Please give some more information on P-SoOP in the manuscript, including when it is planned to be launched and if the track width matches your synthetic assumption. What is the expected accuracy of P-band based SWE estimates?
We cited Yueh et al., 2021 to provide more information on P-SoOP, which is a proposed project that does not yet incorporate snow sensing into the schedule. However, our project was not limited to P-SoOP but is appliable to any intermittent satellite observations. We will make this clearer in the revised MS.

103: Why not also areas in high latitudes?
Good point. We will rewrite this statement to indicate that the method has potential for all snow-covered areas globally.

122: The F2022 snow reanalysis dataset should be described in more detail. What is the meteorological input to generate SWE? Are the applied meteorological variables for the F2022 snow reanalysis dataset the same than those you used for your machine learning approaches (is independency given)?
We will add a little more detail, but of course this is a published paper, so we only give a high-level summary and refer the reader to the archival paper for details.

195: Please define how you classified the years in dry, normal and extremely wet. What are the thresholds?
We will add more details in section 3.2.1 as to the classifications.

338 and Figure 3: As you clearly mention, DNN performs best. However, it is worth to mention that RF shows the lowest PEBAS values for the normal and wet years. Also, more clear statements why DNN performs best would help.
Thank you for the suggestion. We will mention that RF performed best in terms of PEBAS in wet and normal years. We will also provide more information and related references as to why DNN (MLP) performs best.

565: Please add some more discussion on the fact that topography plays an important role regarding the 'choice' of the satellite tracks. Does this play a role in Figure 11 as an increase in the number of ground tracks shows outliers in the course of MAE for WY2017 – 3 ground tracks and WY2008 – 5 ground tracks.
We will add more discussion regarding Figure 11 and Figure S9 in section 4.4 and note that topography plays an important role in the performance of TTA SWE transformation.

**References**

Magnusson, J., Gustafsson, D., Hüsler, F., and Jonas, T.: Assimilation of point SWE data into a distributed snow cover model comparing two contrasting methods, Water Resour. Res., 50, 7816–7835, https://doi.org/10.1002/2014WR015302, 2014.

Pflug, J. M. and Lundquist, J. D.: Inferring Distributed Snow Depth by Leveraging Snow Pattern Repeatability: Investigation Using 47 Lidar Observations in the Tuolumne Watershed, Sierra Nevada, California, Water Resour. Res., 56, https://doi.org/10.1029/2020WR027243, 2020.

Schneider, D. and Molotch, N. P.: Real-time estimation of snow water equivalent in the Upper Colorado River Basin using MODIS-based SWE Reconstructions and SNO[TEL] data, Water Resour. Res., 52, 7892–7910, https://doi.org/10.1002/2016WR019067, 2016.

Walker, A. E. and Goodison, B. E.: Discrimination of a wet snow cover using passive microwave satellite data, Ann. Glaciol., 17, 307–311, https://doi.org/10.3189/S026030550001301X, 1993.

Yueh, S. H., Shah, R., Xu, X., Stiles, B., and Bosch-Lluis, X.: A Satellite Synthetic Aperture Radar Concept Using P-Band Signals of Opportunity, IEEE J. Sel. Top. Appl., 14, 2796–2816, https://doi.org/10.1109/JSTARS.2021.3059242, 2021.

---

## Author Response (AR2)

**Response to Reviewers**

**Re: HESS Manuscript egusphere-2022-470: "Estimating spatiotemporally continuous snow water equivalent from intermittent satellite track observations using machine learning methods" by Ma et al.**

The comments from the editor and the reviewers are reproduced below in black and our corresponding responses are in blue.

Thanks for the great points and comments made by the two reviewers, which are entirely constructive and will, we believe improve the manuscript. We provide a point-by-point response below.

Reviewer #1 (Formal Review for Authors (shown to authors)):

Ma et al. investigate the use of machine learning for track-to-area transformation of satellite observations of SWE in swathes that cover a small fraction of a watershed (it is worth noting that point-to-area transformation with horizontal and vertical distance weighting has long been used for assimilation of in situ snow observations in numerical weather prediction). This study is entirely synthetic, but interesting in anticipation of real remote sensing opportunities. Some clarifications and numerous minor corrections are required. Explainability of the ML results is promised, but there is very little physical explanation in the discussion of the input feature sensitivity tests.

Thanks for the general comments on this manuscript. We have now added more physical explanations and related references relative to the input feature sensitivity test, specifically:

1. Added more appropriate citations (Raleigh and Lundquist, 2012; Luce et al., 2014) explaining that winter precipitation in general is more influential than other meteorological forcings in SWE estimation (lines 521 to 522 in the revised MS).
2. Clarified why removing a specific variable would sometimes increase or decrease MAE of SWE estimation (lines 529 to 536 in the revised MS).
3. Improved the explanation of why positive biases of net longwave radiation and biases in precipitation cause large SWE estimation errors in the normal and wet years (lines 561 to 563 in the revised MS).

71
LiDAR can produce continuous SWE maps; the Airborne Snow Observatory does for this very basin.
We have deleted the description of LiDAR in the Introduction (lines 63-65 in the old MS) and demonstrated that the "track-to-area" method is useful for obtaining spatially continuous SWE where SWE observations are only available along ground tracks (lines 100-101 in the revised MS).

76
"TTA transformation could be achieved by leveraging snow pattern repeatability", but that is not what is done here.
Lines 76-80 (in the old MS) are meant to introduce another TTA SWE transformation method based on snow pattern repeatability. We added more references (Magnusson et al., 2014; Schneider and Molotch, 2016) on point-to-area SWE data transformation methods (lines 81-91 in the revised MS) to give a more comprehensive introduction to TTA or point-to-area SWE transformation applications.

253

Is a 50% perturbation in temperature applied to the Kelvin temperature (which would be enormous) or Celsius temperature (which would be meaningless)? A 50% error in air pressure is also unphysical. To be clear, is the perturbation a fixed bias (as described) and not a random error? If so, why does the DNN not just learn the bias?

The unit here should be Kelvin. The 50% error doesn't mean we increase or decrease 50% of the original value. Instead, we added 50% of the difference between the maximum and minimum temperature for a specific pixel during the study period to the original value. The perturbation is a fixed bias. The reason why we added biases rather than random errors to meteorological inputs is that in snow hydrology, systematic SWE estimation biases are the primary source of estimation error, and it mainly comes from the biases of input meteorological forcings in SWE modeling. We added a clarification (lines 271-273 in the revised MS) as to how we perturb errors for the forcing uncertainty analysis.

323

What is intended by saying "a simple ANN … is capable of learning non-linear relationships… while DNN … can learn more complicated relationships"? More complicated than nonlinear?

Our intention here was to point out the superiority of a Multilayer Perceptron (MLP) relative to a Single-layer Perceptron. A single-layer Perceptron is a neural network with only one neuron and can only understand linear relationships between the input and output data, while with Multilayer Perceptron, horizons are expanded and the neural network can have multiple layers of neurons, which are better adapted to more complex patterns (Gardner and Dorling, 1998). We have added some text (lines 339-343 in the revised MS) to indicate that MLP usually performs better than a Single-layer Perceptron.

330

Levenberg-Marquardt is not the cost function, it is the algorithm used to minimize the cost function.

Thanks for pointing out the error. We have modified the description of Levenberg-Marquardt algorithm (lines 346-347 in the revised MS).

Time series of SWE observations would not be available for wet snow after peak SWE.

We leverage the SWE reanalysis dataset (F2022 introduced in Section 2.2) as the synthetic "truth" for evaluating our SWE estimates. It is available over a full water year, whether the snow is wet or not. We have clarified that the synthetic truth from F2022 was available for a full water year during the study period (lines 134 to 135 in the revised MS)

There is no discussion of why removing a particular variable will sometimes increase and sometimes decrease MAE in Figure 8.

We have added texts (lines 529 to 536 in the revised MS) to explain why some meteorological forcings have negligible influence on TTA SWE transformations in specific years so that removing such variables will sometimes increase and sometimes decrease MAE.

The biases in Figure 9 are fractions, not %. Why are there non-zero changes of MAE for zero bias in WY 2015 and 2017?

Thanks for pointing this out; we have changed the X-axis by a factor of 100. The non-zero changes of MAE for zero bias were an error in plotting Figure 9, which has been corrected below and in the revised MS.

[Figure]

Figure 9.

535

I do not follow the argument for why positive longwave radiation errors cause larger errors. Is it that decreasing longwave radiation cannot increase 1 April SWE above the cumulated snowfall, but increasing longwave radiation can decrease SWE all the way to zero?

We have added a short explanation (lines 561 to 563 in the revised MS) as to why positive longwave radiation errors cause larger errors.

The legend "Observation coverage" could be removed from Figure 10 so that it does not overlap the ground tracks (and four decimal places is excessive for the percentage).

Thanks for the suggestion. We have modified Figure 10 (also shown below) and moved the information about observation coverage to the supplement (Table S1).

[Figure]

Figure 10.

**Minor Corrections**

We have carefully proofread the MS and made minor corrections throughout to eliminate the grammatical mistakes noted by the reviewer, see below.

32
"it therefore, mitigates" – delete comma
Revised.

43
"whether large amounts of snow accumulate"
Revised.

52
"spatial, rather than point observations" – delete comma
Revised.

56
"except during periods of precipitation"
Revised.

59
"topographically complex areas etc. (Le et al. 2017),"– delete etc. and comma
Revised.

113
"snow dominated"
Revised.

157
"Each of the experiments used one of four algorithms"
Revised.

221
"We investigated the performance of the SWE TTA estimation"
Revised.

231
"from the closest previous observation day"
Revised.

352
"for all four algorithms"
Revised.

358
"The reason for larger values"
Revised.

372
"for all three years"
Revised.

375
It would make more sense to say that April 1st SWE is highly correlated with cumulative winter precipitation.
Agreed. We changed this sentence to "April 1st SWE is highly correlated with cumulative winter precipitation." (lines 400-401 in the revised MS).

379
"The error range is larger in WY2008 than in the dry year"
Revised.

391
"underestimation tends to occur"
Revised.

449
The information in this first sentence is repeated in the next sentence.
Revised. We deleted the redundant sentence.

451
"(dry, average and wet)"
Revised.

460
Delete "are more obvious"
Revised.

463
"are shown in Fig. 7"
Revised.

472
"becoming more apparent"
Revised.

505
"that provides the most useful information"
Revised.

507
"The dominance of precipitation is most significant"
Revised.

518
"larger biases have larger impacts"
Revised.

525
"so SWE estimates from the network are slightly different"
Revised.

534
"In addition, we propose"
Revised.

580
"is larger than two"
Revised.

581
"if more than two ground tracks pass"
Revised.

603
"more satellite overpasses do not improve the estimates much"
Revised.

621
"can be used"
Revised.

628
"and wet years"
Revised.

634
"The DNN method is the most accurate"
Revised.

635
"and reduction in the training data size"
Revised.

Reviewer #2 (Formal Review for Authors (shown to authors)):

The manuscript 'Estimating spatiotemporally continuous snow water equivalent from intermittent satellite track observations using machine learning methods' by Ma et al. provides an interesting synthetic study regarding SWE observations from potential satellite tracks for track-to-area transformations. In general, I think such studies are important to be prepared for novel satellite missions and their 'real world' applications. The study is carried out in the Upper Toulomne River Basin in California in the Western United States. The authors apply statistic and machine learning techniques to answer the following four research questions:

(1) How does the spatially distributed April 1st SWE inferred from TTA compare with the synthetic truth, and how do their differences vary in dry, normal, and wet years?

(2) What are the dominant variables for the April 1st SWE estimation in statistical and machine learning TTA methods, and which method has the highest accuracy?

(3) How does the accuracy of the domain-wide SWE estimates from TTA approaches evolve within a season at different temporal observation resolution?

(4) How does the performance of TTA change as a function of the spatial sampling density (number of hypothetical ground tracks), and what is the preferred number of tracks?

The manuscript is well written and organized. I agree with all points raised by reviewer 1. In addition, I have some further points, which need clarification and/or additional information before publication.

Please change the unit of SWE in mm (instead of m or cm) throughout the paper (text, figures, tables), as mm is the unit for SWE.
We have changed the unit of SWE from m to mm throughout the paper as suggested, including text, figures, and tables.

In general, please discuss if the pixel size and satellite track width of approx. 1 km is suitable to accurately describe the quantity of snow and SWE accumulation and ablation within the catchment (and, also for which applications is this resolution sufficient?). Could there be any limitations and need for higher spatial resolutions?
Thank you for the suggestions. The pixel size is 480m and the track width is about 1 km (lines 135-137 in the revised MS). We did not design the parameters but just used the best available data. The synthetic tracks are consistent with a notional design for a P-band signal of opportunity proposed mission (by NASA's Jet Propulsion Laboratory; see Yueh et al., 2021 for more information). The "truth" for the analysis of SWE estimates is the snow reanalysis data (F2022), which is at a 480 m spatial resolution. Designing a satellite with the finer spatial resolution is expected to increase the training size of ML-based TTA SWE transformation, which may lead to better performance of continuous SWE estimation, but about 1 km seems to be what is now technically feasible. Due to space limitations, we only explored one SWE spatial resolution, and we preferred to defer analysis of the sensitivity of ML performance to observation resolution to future studies. We have added more discussion as to what the further steps would be if we would like to consider SWE observations with diverse spatial resolutions (lines 678-683 in the revised MS).

Title: The title is a bit misleading. The study is fully synthetic, but the title 'promises' somehow 'real' satellite track observations. I would recommend to at least include the word 'synthetic' (or 'hypothetic') in the title. Moreover, as you use statistic and machine learning methods it would make sense to add also statistic in the title. Suggestion:

'Estimating spatiotemporally continuous snow water equivalent from intermittent synthetic satellite track observations using statistic and machine learning methods'.

Thank you for your suggestion. We have added "synthetic" to the title to avoid implying that the observations are real. We found that the other qualifiers make the title too cumbersome, and instead, we have changed the title to "Estimating spatiotemporally continuous snow water equivalent from intermittent satellite observations: An evaluation using synthetic data".

Abstract: The abstract should be revised: Please add information on the potential future satellite mission (P-SoOP), on which you relate your work on TTA, and give information on the spatial resolution of your synthetic satellite tracks.

Thank you for your suggestions. What is a bit tricky is that some of the technical background is in a NASA proposal under consideration, and we're limited in what we can say, On the other hand, the Yueh et al. (2021) paper, which we cite, can be leveraged for more information, including some on P-SoOP. We now mention P-SoOP briefly in the abstract (lines 15-17 in the revised MS) as the starting point for our synthetic SWE observations and provided more information on P-SoOP in the introduction part (lines 93-101 in the revised MS).

38+: I am missing a solid literature review
  a) on satellite based SWE and snow height derivation. This should at least include the following references:

  Lievens, H., Brangers, I., Marshall, H.P., Jonas, T., Olefs, M. and De Lannoy, G., 2022. Sentinel-1 snow depth retrieval at sub-kilometer resolution over the European Alps. The Cryosphere, 16(1), pp.159-177.

  Deschamps-Berger, C., Gascoin, S., Berthier, E., Deems, J., Gutmann, E., Dehecq, A., Shean, D. and Dumont, M., 2020. Snow depth mapping from stereo satellite imagery in mountainous terrain: evaluation using airborne laser-scanning data. The Cryosphere, 14(9), pp.2925-2940.

  Thank you for your suggestions on the references. We have added these citations (Lievens et al., 2022; Deschamps-Berger et al., 2020) about snow depth derivation in the introduction part (lines 65-67 in the revised MS).

  b) on other TTA or point-based methods.
  In addition to the TTA and point-based method using snow pattern repeatability (Pflug and Lundquist, 2020), we have added two other papers (Magnusson et al., 2014; Schneider and Molotch, 2016) that discuss other TTA or point-based methods (lines 81-91 in the revised MS).

57: Please add 'wet snow' as further limitation.
We have added a comment on this limitation of PM-based SWE remote sensing (lines 57-61 in the revised MS), with a reference to Walker and Goodison (1993).

82: Please give some more information on P-SoOP in the manuscript, including when it is planned to be launched and if the track width matches your synthetic assumption. What is the expected accuracy of P-band based SWE estimates?
We cited Yueh et al. (2021) and a proposed NASA's SNoOPI satellite (SigNals of Opportunity: P-band Investigation) to provide more information on P-SoOP (lines 93-101 in the revised MS), none of which are yet operational. However, our project was not limited to P-SoOP but was applicable to any intermittent satellite observations.

103: Why not also areas in high latitudes?

Good point. We have rewritten this statement to indicate that the method has potential for all snow-covered areas globally (lines 114-115 in the revised MS).

122: The F2022 snow reanalysis dataset should be described in more detail. What is the meteorological input to generate SWE? Are the applied meteorological variables for the F2022 snow reanalysis dataset the same than those you used for your machine learning approaches (is independency given)?

We have added more details, but this is a published paper, so we only give a high-level summary and refer the reader to the archival paper for details. We have clarified that the ML training samples and domain-wide model inputs are from the same data source (lines 150-152 in the revised MS).

195: Please define how you classified the years in dry, normal and extremely wet. What are the thresholds?

We have added more details explaining how we defined the extremely dry, normal, and extraordinary wet years (lines 206-209 in the revised MS) and how we chose the 12 typical water years (lines 214-219 in the revised MS).

338 and Figure 3: As you clearly mention, DNN performs best. However, it is worth to mention that RF shows the lowest PEBAS values for the normal and wet years. Also, more clear statements why DNN performs best would help.

Thank you for the suggestion. We have mentioned that RF performed best in terms of PEBAS in wet and normal years. Also, we have provided more information and related citations (Segal, 2004) as to why DNN (MLP) performs best (lines 355-365 in the revised MS).

565: Please add some more discussion on the fact that topography plays an important role regarding the 'choice' of the satellite tracks. Does this play a role in Figure 11 as an increase in the number of ground tracks shows outliers in the course of MAE for WY2017 – 3 ground tracks and WY2008 – 5 ground tracks.

We have added more discussion regarding Figure 11 and Figure S7 explaining that topography plays an important role in the performance of TTA SWE transformation (lines 611-617 in the revised MS).

**Reference**

Deschamps-Berger, C., Gascoin, S., Berthier, E., Deems, J., Gutmann, E., Dehecq, A., Shean, D., and Dumont, M.: Snow depth mapping from stereo satellite imagery in mountainous terrain: evaluation using airborne laser-scanning data, The Cryosphere, 14, 2925–2940, https://doi.org/10.5194/tc-14-2925-2020, 2020.

Lievens, H., Brangers, I., Marshall, H.-P., Jonas, T., Olefs, M., and De Lannoy, G.: Sentinel-1 snow depth retrieval at sub-kilometer resolution over the European Alps, The Cryosphere, 16, 159–177, https://doi.org/10.5194/tc-16-159-2022, 2022.

Luce, C. H., Lopez-Burgos, V., and Holden, Z.: Sensitivity of snowpack storage to precipitation and temperature using spatial and temporal analog models, Water Resour. Res., 50, 9447–9462, https://doi.org/10.1002/2013WR014844, 2014.

Magnusson, J., Gustafsson, D., Hüsler, F., and Jonas, T.: Assimilation of point SWE data into a distributed snow cover model comparing two contrasting methods, Water Resour. Res., 50, 7816–7835, https://doi.org/10.1002/2014WR015302, 2014.

Pflug, J. M. and Lundquist, J. D.: Inferring Distributed Snow Depth by Leveraging Snow Pattern Repeatability: Investigation Using 47 Lidar Observations in the Tuolumne Watershed, Sierra Nevada, California, Water Resour. Res., 56, https://doi.org/10.1029/2020WR027243, 2020.

Raleigh, M. S. and Lundquist, J. D.: Comparing and combining SWE estimates from the SNOW-17 model using PRISM and SWE reconstruction, Water Resour. Res., 48, https://doi.org/10.1029/2011WR010542, 2012.

Schneider, D. and Molotch, N. P.: Real-time estimation of snow water equivalent in the Upper Colorado River Basin using MODIS-based SWE Reconstructions and SNO$^{TEL}$ data, Water Resour. Res., 52, 7892–7910, https://doi.org/10.1002/2016WR019067, 2016.

Segal, M. R.: Machine Learning Benchmarks and Random Forest Regression, 2004.

Walker, A. E. and Goodison, B. E.: Discrimination of a wet snow cover using passive microwave satellite data, Ann. Glaciol., 17, 307–311, https://doi.org/10.3189/S026030550001301X, 1993.

Yueh, S. H., Shah, R., Xu, X., Stiles, B., and Bosch-Lluis, X.: A Satellite Synthetic Aperture Radar Concept Using P-Band Signals of Opportunity, IEEE J. SEL. TOP. APPL., 14, 2796–2816, https://doi.org/10.1109/JSTARS.2021.3059242, 2021.